# Developing a framework for the assessment of current and future flood risk in Venice, Italy

Julius Schlumberger[1,4], Christian Ferrarin[2], Sebastiaan N. Jonkman[1], Manuel Andres Diaz Loaiza[1,5], Alessandro Antonini[1], and Sandra Fatorić[3]

[1]Department of Hydraulic Engineering, Faculty of Civil Engineering & Geosciences, Delft University of Technology, Delft, the Netherlands
[2]CNR - National Research Council of Italy, ISMAR - Marine Sciences Institute, Castello 2737/F, 30122, Venezia, Italy
[3]Faculty of Architecture and the Built Environment, Delft University of Technology, Delft, the Netherlands
[4]Deltares, Boussinesqweg 1, 2629 HV Delft, the Netherlands
[5]JBA Consulting, St Philip's Courtyard, B46 3AD, Brimingham, United Kingdom

**Correspondence:** J. Schlumberger (j.schlumberger@posteo.de)

**Abstract.** Flooding causes serious impacts to the old-town of Venice, its residents and its cultural heritage. Despite this existence-defining condition, limited scientific knowledge on flood risk of the old-town of Venice is available to support decisions to mitigate existing and future flood impacts. Therefore, this study proposes a risk assessment framework to provide a methodical and flexible instrument for decision-making for flood risk management in Venice. We first use a state-of-the-art hydrodynamic urban model to identify the hazard characteristics inside the city of Venice. Exposure, vulnerability, and corresponding damages are then modelled by a multi-parametric, micro-scale damage model which is adapted to the specific context of Venice with its dense urban structure and high risk awareness. Furthermore, a set of individual protection scenarios is implemented to account for possible variability of flood preparedness of the residents. This developed risk assessment framework was tested for the flood event of 12 November 2019, proving able to reproduce flood characteristics and resulting damages well. A scenario analysis based on a meteorological event like 12 November 2019 was conducted to derive flood damage estimates for the year 2060 for a set of sea level rise scenarios in combination with a (partially) functioning storm surge barrier, the Modulo Sperimentale Elettromeccanico (MOSE). The analysis suggests that a functioning MOSE barrier could prevent flood damages for the considered storm event and sea level scenarios almost entirely. A partially closed MOSE barrier (open Lido inlet) could reduce the damages by up to 34% for optimistic sea level rise prognoses. However, damages could be 10% to 600% higher in 2060 compared to 2019 for a partial closure of the storm surge barrier, depending on different levels of individual protection.

## 1 Introduction

Flood events are among the most disastrous natural catastrophes, causing significant damages and fatalities all around the world. In Europe, coastal flood events are estimated to affect more than 100,000 citizens, causing losses of about EUR 1.4 billion annually (Vousdoukas et al., 2020). Under consideration of climate change scenarios, future flood damages are expected

to increase due rising sea level (Hinkel et al., 2014).

In this context, hazard and flood risk assessment has been broadly implemented according to the 60/2007/EC directive in the EU (European Commission, 2007). According to the IPCC, flood risk is defined as the combination of a specific hazardous flood event, elements (i.e. infrastructure, people, livelihoods, environment, and cultural, social and economic assets) which might be exposed to a hazard in a certain area, and the vulnerability of these elements, meaning predisposition to be adversely affected (Chen et al., 2021; Cardona et al., 2012). As such, outcomes of a flood risk assessment framework can support systemic and individual decisions to mitigate flood damages or adapt accordingly, increasing preparedness and strengthening coping capacities (Arrighi et al., 2018b; Molinari and Scorzini, 2017; Scorzini and Frank, 2017; Amadio et al., 2016; Thieken et al.; Merz and Thieken, 2009).

A flood risk assessment framework typically follows four steps: 1) hazard modelling, 2) assessment of vulnerability of exposed assets, 3) damage estimation and 4) flood risk estimation (Arrighi et al., 2018a). The application of 2D hydrodynamic models is currently the state of the art method for deriving information about coastal and urban flood events (Yin et al., 2020; Sai et al., 2020; Xing et al., 2019; Teng et al., 2017; Gallien et al., 2014). Damage modelling traditionally focuses on direct, tangible damages in terms of replacement costs related to structures, interiors, and public infrastructure since the cost-benefit analysis of flood mitigation measures is straight forward and indisputable (Molinaroli et al., 2018; Scorzini and Frank, 2017; Dottori et al., 2016; Merz and Thieken, 2009). The vulnerability of exposed assets is determined not only by the type of exposed structure, its construction material (quality), its age, and its level of maintenance (Huijbregts et al., 2014; Drdácký, 2010; Merz and Thieken, 2009), but also by the level of present awareness. Risk awareness influences the level of preparedness by means of physical measures (e.g. permanent or mobile water barriers, emergency works like sand bags) or behavioral adjustments (e.g. adapting the vertical distribution of goods and values). Vulnerability therefore varies highly spatially and temporally (Hudson et al., 2016; Kreibich et al., 2011; López-Marrero, 2010).

This study focuses on the assessment of flood damage in Venice. The low-lying historic city has a long record of flood events (Battistin and Canestrelli, 2006) which is likely to extend into the future mainly due to relative sea level rise and continuing subsidence (Lionello et al., 2021; Međugorac et al., 2020; Morucci et al., 2020; Tiggeloven et al., 2020; Jordà et al., 2012). Since 1987, the city of Venice has been part of the UNESCO World Cultural and Natural Heritage site that spans the Venetian lagoon (Molinaroli et al., 2018). Consequently, not only economic and individual risk prevails, but also risk of damage or loss of highly valued cultural sites. This is expected to contribute significantly to the tangible damages due to special restoration and reconstruction requirements (Arrighi et al., 2018a). Additionally, intangible damages to cultural heritage sites (e.g. loss of historic books or documents, damage to iconic paintings) and their meaning for the cultural identity of the region and nation can be expected (Wang, 2015; Arrighi et al., 2018a).

Thus, dealing with flooding and mitigating adverse effects is an existence-defining task in Venice now and in the future. Over the past decades, flood protection mainly relied on individual preparedness, which was supported by forecasting systems for storm surges incorporated into a multi-stage warning system (Umgiesser et al., 2021; Comune di Venezia., 2016). As part of an extensive flood protection plan, the Modulo Sperimentale Elettromeccanico (MOSE) barrier has been designed following the record flooding in 1966. It is expected to be functional by the end of 2021. The barrier consists of a series of submersed

gates located in the three inlets of the Venetian Lagoon. MOSE is designed to protect Venice against high water exceeding 1.1 m of the local datum Zero Mareographic of Punta della Salute (ZMPS)[1], up to a water level of 3.0 m ZMPS (Cavallaro et al., 2017; Umgiesser and Matticchio, 2006).

Despite much attention to flooding in the city of Venice, no detailed and methodical risk assessment framework is publicly

available. Lack of such a framework makes it more difficult to compare and evaluate various measures (such as the MOSE barrier) and justify the distribution of resources for flood risk mitigation measures (Arrighi et al., 2018a). Moreover, only a few studies on damage or loss modelling cover the old-town of Venice. Some studies investigated potential flood damages based on basic depth-damage relations to analyze the benefit of a functioning barrier (Fontini et al., 2008; Nunes et al., 2005), while others looked into remaining flood risk for floods up to a level of 1.10 m ZMPS (Caporin and Fontini, 2014). These studies

mainly focus on different closure scenarios of the MOSE barrier and consider flood risk implicitly by using a maximum safeguard water level at the city of Venice (Umgiesser, 2020; Cavallaro et al., 2017; Umgiesser and Matticchio, 2006). As such, no risk assessment framework is accessible that captures the flood dynamics or allow for a comprehensive adjustment of exposure and vulnerability due to urban developments for potential long-term use of such frameworks. Flood dynamics might be altered in future because of the operation of the MOSE barrier influencing the bathymetry and thus hydrodynamics of floods in the

Venetian lagoon (Tognin et al., 2022).

The paper proposes a methodical and flexible assessment framework for Venice that is useful to analyze existing and future flood damages for different meteorological storm events. It is methodical, as it uses a hydrodynamic model along with a damage model that can resolve physical damage modelling of separate building components. The framework is flexible in

that both models can be refined to consider additional elements of influence or additional elements at risk. This could be of particular interest for accounting more specific conditions of cultural heritage as well as incorporating additional knowledge about (changing) flood protection measures in Venice. The framework is tested using the second highest recorded flood event for which damage claim data have been collected and made available by the municipality. Those most-recent damage claim data were used to analyse and discuss the suitability of the framework by comparing these empirical data with the simulated

flood damages of the framework.

---

[1]If not highlighted otherwise, all levels refer to the local chart datum in Venice, given as Zero Mareographic of Punta della Salute (ZMPS), corresponding to the mean sea level of the 1885-1909 period. Present mean sea level (2019 annual mean sea level) is today 0.34 m ZMPS.

## 2  Methods

To develop a better understanding of existing and future risk due to damages to structures and cultural heritage in Venice, a risk assessment framework is developed in this study as shown in Fig. 1. High resolution flood hazard characteristics are computed by means of a 2D-hydrodynamic model. They feed into a micro-scale damage model to estimate expected absolute direct damages of the exposed buildings (Dottori et al., 2016). The flood model is calibrated and partly validated using data from the storm surge of 12 November 2019. Additionally, a damage claim data-set for the the same event is used for performance analysis of the damage model. Finally, the framework is applied to a set of scenarios of varying sea level change and MOSE closure to analyze potential developments of flood damage instead of flood risk in the mid-term future. This simplification was used as information about (future development of) return periods of the studied storm surge event, and probabilities of barrier failure scenarios are not available. However, the derived development of flood damage estimates as provided in this study can be easily translated into flood risk information by accounting for the probabilistic information.

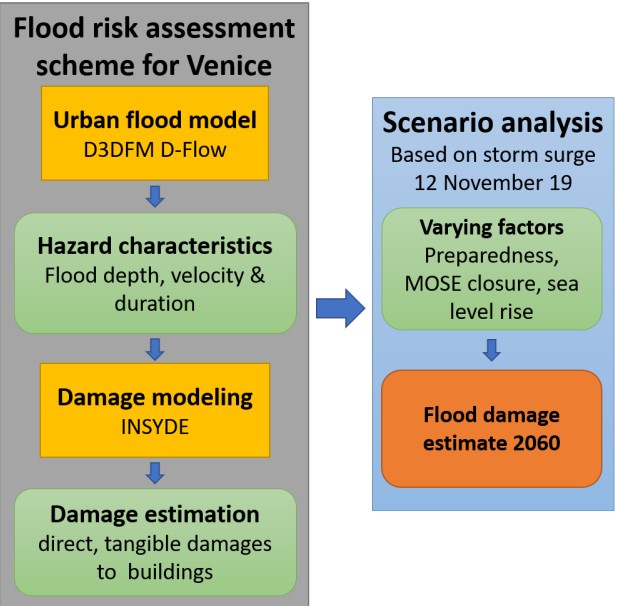

**Figure 1.** Risk assessment framework

### 2.1  Study area and storm event of 12 November 2019

The old-town of Venice covers an area of about 6 $km^2$ and is pervaded by more than 100 canals of depths between 1 and 5 meters (Madricardo et al., 2017). The old-town is located in the Venetian lagoon, the largest in the Mediterranean with an area of about 550 $km^2$. The lagoon is connected to the Northern Adriatic Sea via three inlets at Lido, Malamocco and Chioggia,

see Fig. 2.

On 12 November 2019, the second highest sea level since the beginning of measurements (1872) flooded the old-town of Venice and other parts of the Venetian lagoon. The maximum measured water level inside the old town was 1.89 m ZMPS, measured by the tidal gauge station Punta della Salute at 22:50 on 12 November 2019. It was comprised of a tidal contribution of 0.36 m, 0.47 m of storm surge induced by a strong Sirocco wind over the Adriatic Sea, 0.35 m of long-term preconditioning, and 0.34 m mean sea level with regards to the local datum (Ferrarin et al., 2021). At the same time, a secondary, local cyclone passed over the Northern Adriatic Sea resulting in additional set-up by causing an inverse barotropic effect and very high wind speeds from south-westerly directions of about 70 up to 110 km/h. It is noteworthy that the secondary low pressure field was not forecasted properly which lead to an underestimation of the flood by about 0.40 m (Ferrarin et al., 2021). Unlike a storm event that occurred in 2018 where an even higher tidal peak (1.56 m ZMPS) coincided with low astronomical tides (-0.10 m ZMPS), the extreme sea level of 12 November 2019 was the product of less extreme, thus more likely conditions (Morucci et al., 2020; Cavaleri et al., 2019).

As a response to the unexpected extreme meteorological event of 12 November 2019, financial support to the affected parties was provided in two rounds: 1) limited amounts for immediate response (up to EUR 5,000 for residents and EUR 20,000 for non-residential entities (companies, NGOs,...)) and 2) support for more extensive flood damages. Residents and entities could apply for compensation for either one or both rounds. In total, 7,644 eligible claims were issued inside the study area with a total cost of EUR 56.2 million[2].

For residents and entities that submitted only immediate response claims (3,728 claims covering EUR 26.99 million of damages), physical addresses of the claimants are publicly available. It was possible to allocate 95% of the reported immediate response claims (EUR 25.73 million) to 2,778 structures inside Venice using a set of 33,096 addresses[3]. For claimants that submitted claims in both rounds or just for more extensive flood damages (EUR 29.21 million), the available information provided was aggregated by city-district for data protection reasons[4].

## 2.2 SLR and MOSE scenarios

The developed framework is applied to a set of seven different scenarios to derive indications of potential development of flood damage and flood risk in future. The scenarios differ in mean sea level and closure behaviour of the MOSE barrier as summarized in Tab. 1. For all scenarios, the meteorological forcing of a storm equivalent to the extreme event of 12 November 2019 is used. SLR0 considers a mean sea level as present in 2019. 'SLR0-allopen' represents the real flood event of 2019

---

[2]Data made available by the Office of the Delegated Commissioner for the management of exceptional meteorological events from 12 November 2019 in the territory of the Municipality of Venice.

[3]accessed at: https://portale.comune.venezia.it/node/117/12181978

[4]More information on and analysis of the available damage claim data can be found in the supplementary material of this study.

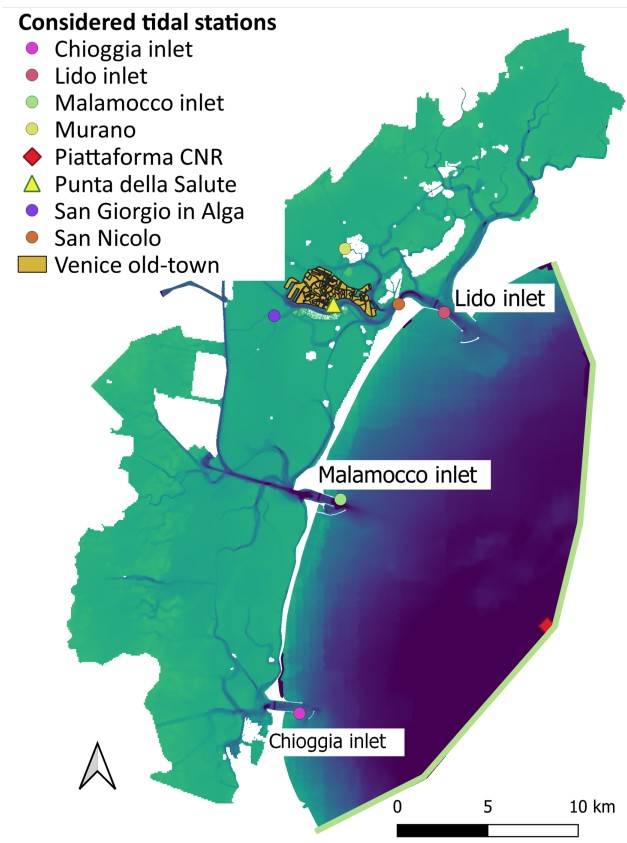

**Figure 2.** Study area consisting of part of the Adriatic shelf, the Venetian lagoon and the old-town of Venice. Green line indicates applied boundary condition for the water-level time-series.

without an operational MOSE barrier. Scenarios of 0.15 m and and 0.45 m sea level rise with respect to 2019 are selected in line with latest research on sea level rise prognosis in Venice. They correspond to the lower and upper confidence bounds of the projected sea level change in the Northern Adriatic Sea under RCP2.6 and RCP8.5 scenarios for the year 2060, respectively (Zanchettin et al., 2021). Regarding the MOSE barrier, two closure states are considered: a fully functioning MOSE barrier ('allclosed') and a set-up where all inlets except for the Lido inlet close ('lidoopen'). Previous works (Mooyaart and Jonkman, 2017; Vrancken et al., 2008) and experiences from practice in Venice (Colamussi, 1992; Umgiesser and Matticchio, 2006) have shown that there is a probability of non-closure of storm surge barriers. In an a-priori assessment of the inlets with regards to their dimensions and proximity to the old-town of Venice, we identified that non-closure of the Lido inlet ('lidoopen') is likely the most critical partial-closure scenario. This choice in line with previous studies indicating the prominent importance of this inlet to manage water levels in Venice (Cavallaro et al., 2017; Umgiesser, 2020).

**Table 1.** Applied scenarios to assess future flood damages

| | | scenario | MSL [m ZMPS] |
|---|---|---|---|
| present conditions | | SLR0-allopen | 0.34 |
| | | SLR0-allclosed | 0.34 |
| | | SLR0-lidoopen | 0.34 |
| RCP 2.6 scenario | | SLR1-allclosed | 0.49 |
| | | SLR1-lidoopen | 0.49 |
| RCP 8.5 scenario | | SLR2-allclosed | 0.79 |
| | | SLR2-lidoopen | 0.79 |

## 2.3 The modelling framework

As visualized in Fig. 1, the modelling framework consists of a combination of a hydrodynamic and a damage model which is presented in this section.

### 2.3.1 Hydrodynamic model

In the study area, hydrodynamic models have been used frequently but do not account for the urban area of Venice (Umgiesser et al., 2021; Ferrarin et al., 2015; D'Alpaos and Defina, 2007; Umgiesser et al., 2004; Roland et al., 2009). Studies looking into the distribution of flood depths in Venice have used a static model, also called a bathtub model (Cellerino et al., 1998)[5]. This uses the water level at the tidal gauge of Punta della Salute and compares it with the surface elevation of the old-town of Venice to identify the flood extent and depth. A bathtub model assumes instantaneous flooding, neglecting the process of flood wave progression and therefore possibly overestimating the flood depths inside the city. Using a 2D hydrodynamic model might be able to capture the flood progression into the city, the role of sewage networks and other processes more realistically while also providing the appropriate framework to account for other flood parameters such as flow velocity. Moreover, the hydrodynamic model can be forced with variable water levels at the boundaries of the nested sub-models, thus accounting for strong water level gradients over the city registered by the observations during the 12 November 2019 event.

For this study, a 2D hydrodynamic model based on Delft3D Flexible Mesh Suite 2021.04 was used (Deltares, 2021). The software provides a flexible unstructured grid framework which facilitates the grid generation in the complex coastal and urban setting (Martyr-Koller et al., 2017). Furthermore, it provides additional modules that can be used for a better physical representation of the system. Only 2D flow was considered in this study, but the model allows users to account for additional processes

---

[5]also mentioned here: http://www.comune.venezia.it/maree

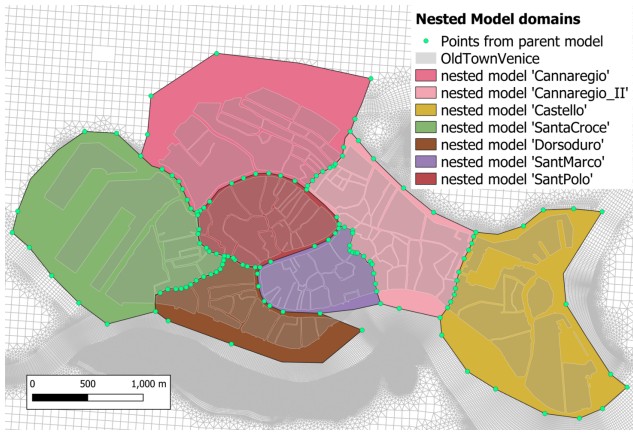

**Figure 3.** Nested model domains with observation points from parent model used as boundary forcing

like wave action or 1D flow of the sewage system[6].


An offline grid nesting framework was chosen, consisting of a parent model covering the study area and seven sub-models of higher resolution covering the area of the old-town of Venice. The parent model used 2.73 million elements covering the study area with an average grid size between 2.6 m in the old-town and 200 m at the Adriatic shelf. In the seven nested models, grid size was increased to an average of 1.3 m to reproduce the narrow street system in Venice. Water level time-series from

the parent model simulation were extracted at 168 locations inside and around the old-town of Venice. Each nested model is enclosed by a sub-set of these locations as shown in Fig. 3. Consequently, for each nested model, the water level time-series of the enclosing locations were used as the boundary inputs driving the hydrodynamic simulation. As such, the sub-models did not exchange information among each other but were run independently. Inconsistencies in flow velocities and water levels due to the lack of interaction between the sub-models were neglected given that most interaction was assumed to occur through the

canals which were sufficiently captured already in the parent model using a resolution of 2.6m within the city. Inconsistencies in flow velocities and water levels due to the lack of interaction between the sub-models were neglected given that most interaction was assumed to occur through the canals which were sufficiently captured already in the parent model using a resolution of 2.6m within the city. Within each nested model, the maximum water level per building was derived by taking into account the maximum water levels of every grid point within a 4m distance from the building perimeter.


Most recent information on the depth of the lagoon flood plains, channels, and the elevation of the islands of the old-town were accessed from various sources. Table 2 presents an overview of all the elevation data used. All altimetry data were corrected to refer to ZMPS, the local chart datum in Venice.

---

[6]A more detailed reasoning along with additional information on the model set up are described in the supplementary material.

**Table 2.** List of used altimetry data

| altimetry data | datum | resolution | year | source |
| --- | --- | --- | --- | --- |
| Venetian lagoon | IGM42 * | 10m | 2002 | Sarretta et al. (2010) |
| Tidal channels | ZMPS | 0.50m | 2013 | Madricardo et al. (2017) |
| Adriatic shelf | LAT ** | 550m | 2018 | EMODnet (2018) |
| old-town surface | IGM42 | 1m | 2011 | ArcGis (2020)*** |
| Canals in old-town | IGM42 | varying | 2000 | City of Venice (2000) |

\* 0 m IGM42 (l'Istituto Geografico Militare Genua 1942) corresponds to + 0.23 m ZMPS

\*\* When analyzing the water level time series of the Aqua Alta platform for different months of 2019, the LAT (Lowest Astronomical Tide) was chosen to correspond approximately to - 0.40 m ZMPS.

\*\*\* The original altimetry data were collected by the RAMSES project (www.ramses.it which was conducted in the year 2011 as a topographic survey characterized by high precision (altimetric of 1 cm and planimetric of 2 cm). The used files have been made available by ArcGIS. Used data were accessed here: https://learn.arcgis.com/en/projects/map-venice-in-2d-and-3d/ (accessed: 08/04/2021)

Constant standard values were used for the viscosity, diffusivity, and density as the flow in the Venetian lagoon is relatively well mixed without stratification (Ferrarin et al., 2010). Roughness was added as Manning-type n. A standard roughness value of 0.023 was applied to the entire study area and eventually altered in different areas of the model domain based on the predominant characteristics, as outlined in Tab. 3. Roughness was used as a calibration factor and checked that the values lie in the range of commonly applied roughness values for the different land types (Ahn et al., 2019; Xing et al., 2019; Ferrarin and

Umgiesser, 2005).

**Table 3.** Applied roughness values

| area | n |
| --- | --- |
| tidal channels | 0.025 |
| tidal plains | 0.040 |
| northern lagoon | 0.020 |
| vegetation Venice | 0.035 |
| streets Venice | 0.019 |
| canals Venice | 0.023 |
| inlets | 0.030 |

Similarly, the wind-induced shear stress, by means of drag coefficient, was used as a calibration parameter. It was implemented based on a linearly increasing relation between wind speed and wind drag developed by Smith and Banke (1975). Notably, their relation was derived for wind speeds between 6 and 21 m/s, but extreme wind speeds for the 12 November 2019

reached up to 27 m/s. Therefore a higher drag coefficient of 0.00876 (for 100 m/s wind speed) was used. A comprehensive analysis of commonly used wind drag formulations confirmed that the chosen drag coefficient is within the range of available

estimates (Bryant and Akbar, 2016). In addition, it was confirmed that the chosen values are in line with other Delft3D-FM studies of the Venetian lagoon[7].

The barrier system was modelled by means of a set of three simple weirs with a crest height defined by a time-series. It is assumed that the barrier crest height increases at constant speed from the bottom of the respective inlet up to a height of 3.00 m ZMPS and closes within 30 minutes (Umgiesser et al., 2021). For the considered meteorological storm conditions, the MOSE barrier starts closing when the tidal gauge station of Punta della Salute reaches a water level of 0.65 m ZMPS (Zampato et al., 2016). This threshold is assumed to be constant for all analyzed scenarios. The starting time of closure was determined by

modelled tidal gauge information from Punta della Salute for the different scenarios without a closing MOSE barrier, see Tab. 4.

**Table 4.** Closure times for scenarios

| Scenario | Closure time |
| --- | --- |
| SLR0 | 12/11/19 18:40 |
| SLR1 | 12/11/19 18:10 |
| SLR2 | 11/11/19 18:10 |

### 2.3.2   Damage Modelling

While general damage drivers are broadly acknowledged (Patt and Jüpner, 2013; Kelman and Spence, 2004), the exact effect of hazard characteristics on an exposed structure is still poorly understood as it also heavily depends on the material and its

quality (Huijbregts et al., 2014; Merz and Thieken, 2009). This is particularly relevant for cultural heritage sites built using materials which have deteriorated by centuries of existence (Drdácký, 2010). Consequently, the chosen model was selected with special care to allow for an inclusion of differing vulnerability characteristics.

    Various approaches and post-flood data analyses have been conducted to understand the relationships between the flood

hazard characteristics and corresponding tangible, direct damages. Several comparative studies have looked into the character-ization and performance analysis of some frequently used damage models (Molinari et al., 2020; Gerl et al., 2016)[8]. In general, loss estimates reflect high uncertainties and disparities because of the inaccuracy of the models and the lack of knowledge about the system in which they have been applied (Scorzini and Frank, 2017; Gerl et al., 2016).

In this study, a flood model based on INSYDE (In-depth Synthetic Model for Flood Damage Estimation) was applied. INSYDE is a synthetic damage model developed based on 'what if' - scenario analysis to provide a methodical and generalized perspective on the flood-damage process for different building components individually (Dottori et al., 2016). It has been

---

[7]Personal communication G.Lemos, 24.05.2021

[8]An overview of commonly applied damage models in Italy can be found here: http://www.fdm.polimi.it/models (accessed 27/04/2021)

validated based on flood data from a river flood in Caldogno, Veneto, 2010. INSYDE is a multi-parametric model adopting 23 parameters to describe hazard, exposure and vulnerability characteristics of buildings[9]. As the model explicitly considers many damage mediating factors, it allows for direct adjustments or extensions of the model based on the available knowledge or considered research purposes (Molinari et al., 2020; Scorzini and Frank, 2017; Dottori et al., 2016). As such, it is ideal to be extended to include new building types, e.g. cultural heritage sites like churches etc., with specific hazard-structure responses. The INSYDE model also makes use of building-type categorization to account for differences in the vulnerability characteristics between typical buildings in a study area. As a result, the absolute damage, D, per structure is calculated as the sum of a set of damage components summarized in Tab. 5:

$$D = \sum_{i=1}^{n}\sum_{j=1}^{m} C_{i,j} = \sum_{i=1}^{n}\sum_{j=1}^{m} up_{i,j} * ext_{i,j} * E[R] \tag{1}$$

where $j$ represents the damage component and $i$ describes the considered activity, e.g. cleaning, removal, and replacing. $up_{i,j}$ is the unit price per damage component for for a given activity, $ext_{i,j}$ is the extent of exposed component and $E[R]$ the (expected) damage ratio. $E[R] \sim [0,1]$ is derived from fragility functions for different hazard characteristics with gradual influence on the damage. They have been developed based on expert knowledge but are transparently reported as part of the supplementary material of Dottori et al. (2016).

These fragility functions follow truncated normal distributions and relate a probability of damage of a specific component to one flood hazard characteristic: flood depth, flood velocity, or flood duration. In the present study, flood depth is the only damage mediating factor since flow velocity and flood duration were found to be too low to add an additional source of damage (Dottori et al., 2016; Penning-Rowsell et al., 2005)[10]. The fragility functions allow not only for a deterministic multi-parametric consideration of the flood-structure interaction, but also to account for uncertainties in the flood-structure interaction in a probabilistic framework. An example is shown in Fig. 4: damage to partition walls occurs if the partition walls absorb too much water to be dried up, i.a. if water depth exceeds a certain threshold (Dottori et al., 2016). The fragility function can be used to determine an expected damage ratio or expected share of damaged partition wall for a given flood depth. However, damage to partition walls due to a certain water depth could range from 'no damage' to 'full damage', depending on factors such as the quality of wall (material). In the probabilistic framework, a large set of realizations for each component is drawn to derive the 5- and 95-percentiles expressing an optimistic and pessimistic estimate of the absolute damages. Even though the probabilistic framework was not used in this study, it may be useful in case of extending the framework to explicitly cover cultural heritage sites in Venice which may be more sensitive to varying flood characteristics.

Information on the individual building area and extent were derived from cadastral data of the city of Venice[11]. A total of 14,460 structures were considered. Information on the structural properties, the year of construction and the maintenance

---

[9]More details regarding the background and set up of the INSYDE model is provided in the supplementary material of this study.

[10]Results of the hydrodynamic model suggest that flood velocities are generally lower than 0.3 m/s and the flood duration is between 2 and 4 hours.

[11]Accessible here: http://geoportale.comune.venezia.it(accessed 05/07/2021)

**Table 5.** Damage components considered in INSYDE. Italic: not taken into account in this study.

| | sub-component | | | sub-component |
|---|---|---|---|---|
| **Clean-up** | C1 – Pumping | | **Structural** | *S1 – Soil consolidation* |
| | C2 – Waste disposal | | | *S2 – Local repair* |
| | C3 – Cleaning | | | *S3 – Pillar repair* |
| | C4 – Dehumidification | | | |
| **Removal** | R1 – Screed | | **Finishing** | F1 – External plaster replacement |
| | *R2 – Pavement* | | | F2 – Internal plaster replacement |
| | R3 – Skirting | | | F3 – External painting |
| | R4 – Partition walls | | | F4 – Internal painting |
| | R5 – Plasterboard | | | *F5 – Pavement replacement* |
| | R6 – External plaster | | | F6 – Skirting replacement |
| | R7 – Internal plaster | | **Windows & Doors** | W1 – Door replacements |
| | R8 – Doors | | | W2 – Window replacements |
| | R9 – Windows | | | |
| | R10 – Boiler | | | |
| **Non-structural** | N1 – Partition replacements | | **Building systems** | P1 – Boiler replacement |
| | N2 – Screed replacement | | | P2 – Radiator painting |
| | N3 – Plasterboard replacement | | | *P3 – Underfl. heating replacement* |
| | | | | P4 – Electrical system replacement |
| | | | | P5 – Plumbing system replacement |

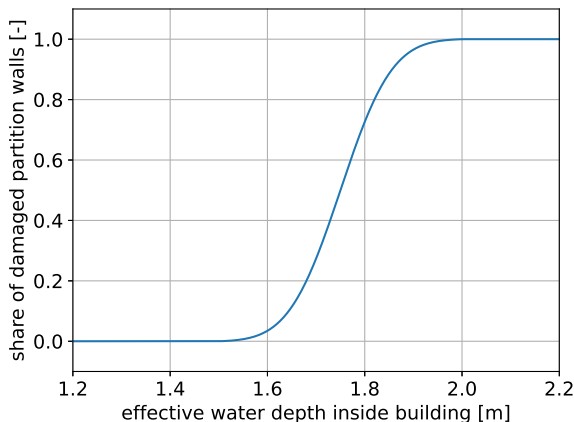

**Figure 4.** Fragility function for partition walls relative to water depth

level were accessed from census data from year 2011 by the Italian National Institute of Statistics (ISTAT, 2020). The census data is not building-specific but aggregated in census blocks covering multiple buildings. As a consequence, the most frequent characteristic was applied to all buildings within a census block[12].

GoogleMaps StreetView was used to gather visual information about typical house fronts, size and number of windows along with information about possible elevations of the entrance at ten random locations in different districts of the old-town. At each of the random locations, we regarded house fronts on both sides up to a distance of 50 to 250m in various directions from the starting point. In this way, we obtained information regarding an estimate of 300 buildings. Length information were estimated based on expert judgment, available scales (e.g. door dimensions). In this way, a first-order estimation of building information was obtained in absence of available statistical data. These building characteristics were confirmed with local inhabitants. Moreover, advertisements by real estate agencies were used to characterise the interior of housings on the ground-floor in the old-town of Venice. They were used to estimate the average minimum height of electrical sockets, type of floor cover, presence of water-proof skirting boards and other protection measures. In addition, graphic documentation of the 12 November 2019 storm surge by the Aqua Grande project[13] was used to search for installed flood protection measures.

The typical characteristics of residential buildings were found not to differ significantly from the implemented characteristics in INSYDE. One major difference related to the external wall perimeter exposed to floods was detected and incorporated as a new parameter $EP_{eff}$: most buildings in Venice are attached to other buildings reducing the exposed perimeter. Additionally, a new building type 'buildings with economic activities on the ground floor' (BEA) was added to account for observed differences in the vulnerability characteristics from typical residential buildings: the windows are generally larger (increased from 1.4m x 1.4m to 2m x 2m), the window sills are lower (new sill height of 0.5m instead of 1.2m), and many shops are on ground level without any steps of elevation. Additionally, the internal perimeter (reduced from 2.5 to 1.5 time the external perimeter) and number of doors are smaller (reduced to 3 per 100 $m^2$).

It was detected that many buildings had installed mobile protection systems, mainly bulkhead protections, at doors and windows to protect the interior from flooding during the 12 November 2019 storm event. Other protection measures were not commonly installed and therefore not incorporated in the damage model. A new parameter, $BuHe$ representing the bulkhead protection height, was implemented to mediate the water level inside the buildings. Due to lack of data on the spatial distribution and protection height of mobile protection systems, three conceptual individual protection scenarios (IPS) were characterized and applied: medium IPS, risk averse IPS and risk-taking IPS. For the risk taking IPS, it was assumed that no bulkhead protection was installed at all. For the medium IPS, it was assumed that residents would install bulkheads protecting their building against the forecasted maximum water level ($FC$) at Punta della Salute incremented by a safety margin of 10 cm. For a risk averse IPS, the protection height also refers to the forecasted maximum water level at Punta della Salute but is

---

[12]More detailed information on the census block data can be found in the supplementary material of this study.

[13]accessed from: https://www.aquagrandainvenice.it/en/welcome

incremented by a safety margin of 50 cm. The water level $h$ inside the buildings is consequently calculated as

$$h = h_e - GL - BuHe \quad \text{and} \quad BuHe = \begin{cases} 0 & \text{, if risk-taking IPS} \\ FC + 0.1 & \text{, if medium IPS} \\ FC + 0.5 & \text{, if risk averse IPS} \end{cases} \tag{2}$$

where $h_e$ is the water level outside the buildings, $GL$ is the ground floor level of the considered structure and $BuHe$ is the bulkhead protection height as visualized in Fig. 5. $FC$ was set to 1.50 m ZMPS for 'SLR0-allopen' and to 1.10 m ZMPS in all other scenarios given that a functional MOSE barrier is expected to keep the water level below a threshold of 1.10 m ZMPS.

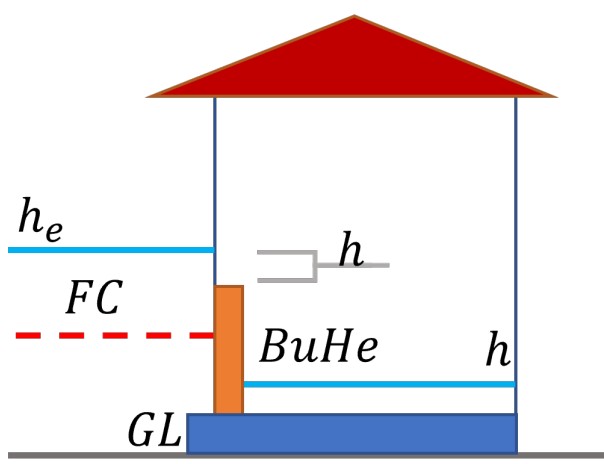

**Figure 5.** Visualization of bulkhead protection height

As a third parameter, information on the cultural heritage status of buildings[14] inside Venice was used to account for higher reconstruction costs. In line with a previous study assuming cost increase of reconstruction for historic buildings by 7 to 11% (Fontini et al., 2008), total damage costs were incremented by 10% in case of cultural heritage status. This is also in line with commonly mentioned ranges of reconstruction costs in Venice[15]. Unit prices for cleaning, removal, and replacement were used from the INSYDE model assuming that those values do not significantly vary across Italy. INSYDE provides prices at 2015 price level. They were corrected for inflation and referenced to the year 2019.

---

[14]Provided by the cultural heritage office of the city of Venice.

[15]See for example here: http://costo-ristrutturazione-casa.it/costo-ristrutturazione-appartamento-venezia/ (accessed 09/04/2021)

# 3 Results

This study developed a methodical framework to assess present and future flood risk in the historic city of Venice. As such, a hydrodynamic model was developed, calibrated and validated. In addition, a damage model was compared against available damage claim data of the storm event of 12 November 2019. Ultimately, the framework was applied to analyze development of future flood damages under sea level rise scenarios in case of a (partially) closing MOSE barrier.

## 3.1 Calibration & validation of the hydrodynamic model

For calibration and validation of the hydrodynamic parent model, modelled water levels were compared against measurements obtained at seven tidal gauge stations: Lido inlet, Malamocco inlet, Chioggia inlet and San Nicolo, Murano, San Giorgia in Alga and Punta della Salute which are located in close proximity to the old-town, as visualized in Fig. 2. Water level informa- tion was provided by the meteo-tidal network of the Venice Lagoon[16]. Three events were used for calibration and validation purposes as shown in Tab. 6. For the tide calibration, a summer period was chosen where influence of wind on the water levels inside the lagoon can be expected to be low. The full model was calibrated for the storm event of 12 November 2019 and finally validated for another storm event from October 2018.

**Table 6.** Considered conditions for calibration and validation

| used for | period |
| --- | --- |
| tide calibration | 01/07/13 00:00 - 04/07/13 23:50 |
| wind calibration | 12/11/19 00:00 - 13/11/19 02:00 |
| model validation | 28/10/18 16:00 - 30/10/18 02:00 |

To evaluate the performance of the model, the Pearson R coefficient and the Root-Mean-Square-Error (RMSE) were used. Results for the three runs are compiled in Tab. 7 and suggest that measured data can be reproduced well, including the storm surge peaks for the wind calibration and validation run. Accuracy of the maximum flood peak lies within a margin of $\pm 5 cm$. For San Nicolo, Malamocco and Murano, the observed water level data were partly corrupted or not available[17].

The nested models were used to derive the flood depth estimates inside the city. Analysis of the difference in water depth estimates inside the old-town of Venice from the parent and nested model domains suggest that the grid resolution of the hy- drodynamic model has significant impact on the flood characteristics inside the city. As Fig. 6b shows, a coarser grid tends to provide lower flood depth estimates. A coarser grid may fail (more often) to resolve possible flow paths in the very narrow

---

[16]accessed here: https://www.venezia.isprambiente.it/rete-meteo-mareografica

[17]Further analysis of the results can be found in the supplementary material of this study.

**Table 7.** Parent model performance

| station | tide calibration | | wind calibration | | model validation | |
| --- | --- | --- | --- | --- | --- | --- |
| | R | RMSE [m] | R | RMSE [m] | R | RMSE [m] |
| Murano | 0.969 | 0.048 | - | - | 0.992 | 0.078 |
| PuntaSalute | 0.977 | 0.043 | 0.987 | 0.078 | 0.990 | 0.068 |
| SanGiorgio | 0.970 | 0.049 | 0.989 | 0.070 | 0.989 | 0.097 |
| SanNicolo | 0.989 | 0.027 | 0.945 | 0.136 | - | - |
| Malamocco | 0.971 | 0.054 | 0.984 | 0.081 | - | - |
| Chioggia | 0.993 | 0.025 | 0.977 | 0.091 | 0.934 | 0.114 |
| Lido | 0.986 | 0.040 | 0.974 | 0.097 | 0.945 | 0.121 |

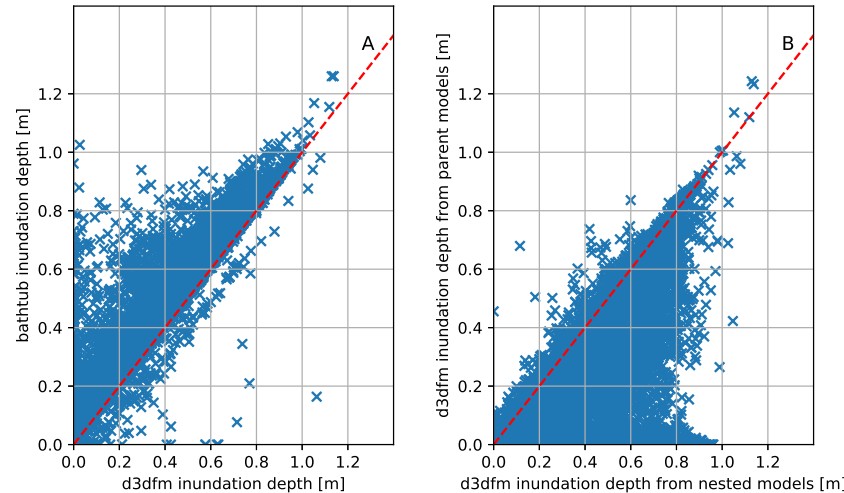

**Figure 6.** Average flood depth estimates of buildings for old-town of Venice (excluding buildings in nested model 'Castello' (see Fig. 3)). A: Cross-model comparison between bathtub and d3dfm (grid-resolution of 1.3m). B: Comparison of flood depth estimates for different grid resolutions of hydrodynamic model (y-axis: grid-resolution of 2.6m; x-axis: grid-resolution of 1.3m).

street-system in Venice limiting water flow into the old-town.


Calibration was not possible inside the old-town due to lack of available measured data. Instead, a cross-model comparison of the nested model flood depth estimates with a simple bathtub model was used to analyze the average maximum flood depth estimates for the 12 November 2019 storm event. The bathtub model tends to provide higher inundation estimates, as shown in Fig. 6. Additionally, the hydrodynamic model gives high flood depths for some buildings while the bathtub models suggests that those structures are not affected by water levels at all (or to a much lesser degree). This unexpected result was linked to grid

instabilities of the nested models. In total, higher water levels were suggested by the hydrodynamic model at 383 buildings. Additionally, grid instabilities of the nested sub-model 'Castello' (refer to Fig. 3) could not be resolved, resulting in missing flood depth data based on the hydrodynamic model for 2,098 buildings (14 % of the total number of buildings). For buildings affected by instabilities, flood depth estimates from the bathtub model were used for the damage modelling of these buildings.


## 3.2 Damage model performance

To analyze the performance of the transferred model, the total modelled damages for the old-town were compared against the total sum of the eligible 7,644 damage claims. Additionally, a structure-wise analysis was conducted for the sub-set of 2,778 structures with 3,728 immediate response claims. A total of 94 immediate response claims (2.5 % of immediate response 340 claims, amounting for 656,264 EUR in claim volume) were located in the sub-model "Castello". As indicated before, we used flood depth estimates from the bathtub model resulting in minor effects on the structure-wise results.

**Table 8.** Comparison of damage claims and estimates based on hydrodynamic (d3dfm) and bathtub (btb) flood depth estimates [EUR million]

| | | | INSYDE | | claims |
|---|---|---|---|---|---|
| | | | d3dfm | btb | |
| sub-set of structures | | risk averse IPS | 12.9 | 13.1 | |
| | | medium IPS | 42.0 | 47.5 | 25.7 |
| | | risk taking IPS | 63.1 | 65.8 | |
| all structures | | risk averse IPS | 52.3 | 53.8 | |
| | | medium IPS | 166.3 | 193.1 | 56.2 |
| | | risk taking IPS | 253.6 | 269.9 | |

As shown in Tab. 8, the damage model is able to reproduce the damage claims well: for both sets of considered structures, reported damage claims fall inside the range of modelled damage estimates for the different IPS. While the total volume of 345 reported immediate response claims corresponds to an individual protection scenario between 'risk averse' and 'medium', the total volume of all reported damages is closer aligned with a risk averse IPS. Furthermore, damage estimates based on the bathtub calculations are generally larger, which is in line with the lower level of flood depth estimations by the hydrodynamic model. The difference increases with decreasing level of individual protection.

Additionally, a structure-wise comparison was conducted for 2,778 structures. As shown in Tab. 9, correlation and average relative error, computed as the ratio of the reported damage and the estimated damage per building, suggest limited alignment of the modelled damages with the reported claims. Both indicators suggest that the damage claims might be slightly better estimated for damages computed based on bathtub flood estimates. Furthermore, claims might be slightly better estimated based on an medium IPS or risk taking IPS for most buildings. At the same time the RMSE, which gives more weight to extreme

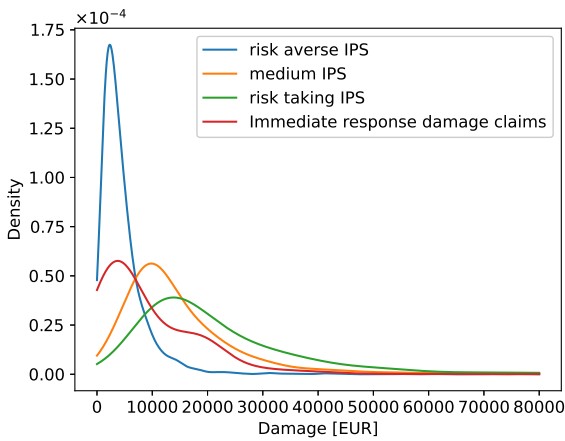

**Figure 7.** Kernel density plot: damage estimates and claims

variations due to its definition, is lower when assuming a risk averse IPS. Moreover, the Kernel density plot gives insight in the relative frequency of damages as shown in Fig. 7. In a risk averse IPS, the number of structures with rather low damages is overestimated, meanwhile larger damages are underestimated. The opposite applies to risk neutral and risk taking scenarios.

**Table 9.** Performance indicators of damage estimates based on hydrodynamic (d3dfm) and bathtub (btb) flood depth estimates for structures with immediate response claims

|  |  | risk averse IPS | medium IPS | risk taking IPS |
|---|---|---|---|---|
| d3dfm | R [-] | 0.22 | 0.26 | 0.26 |
|  | RMSE [EUR] | 19,382 | 22,158 | 29,332 |
|  | RE [%] | 308.9 | 87.8 | 55.5 |
| btb | R [-] | 0.22 | 0.25 | 0.26 |
|  | RMSE [EUR] | 19,384 | 23,298 | 30,122 |
|  | RE [%] | 304.9 | 71.5 | 51.8 |

      According to the INSYDE model, the most affected building components are external and internal plaster removal (R6, R7),
replacement (F1, F2) and painting (F3, F4), followed by costs for the replacement of electrical (P3) and plumbing systems (P4), as shown in Fig. 8. The model often suggests no damage for many damage components as hazard characteristics are below thresholds for which damage is reported to occur. It can be seen that the medium IPS leads to limited damage reduction regarding plaster, but a strong reduction for the building systems. In a risk averse IPS, no damage occurs inside the buildings.

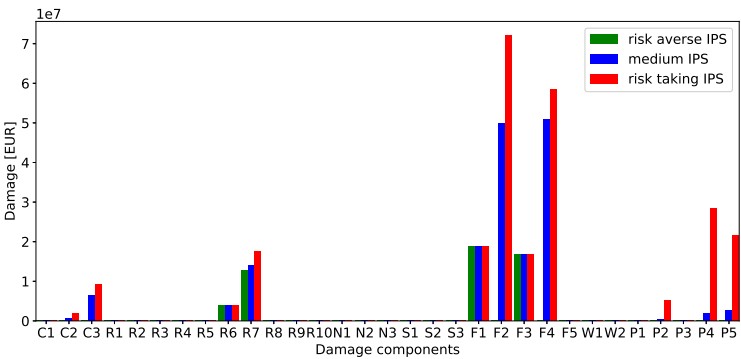

**Figure 8.** Damage components and damage estimation for all structures for SLR0-allopen

It is worth mentioning that damage estimates based on flood depth information from the bathtub model generally give similar damage estimates for both sets of considered structures; deviations for risk averse and risk taking IPS sit between 1.5 and 6.3%. For the medium IPS, damages are about 13.1 to 16% higher when using bathtub model depth estimates. This is a reasonable observation, given that the bathtub model generally provides higher flood depth estimates. As a result, the number of buildings where the flood depth of the bathtub model exceeds the protection height but flood depth of the hydrodynamic model does

not exceed the protection height is higher for the medium IPS than for the risk taking or risk averse IPS. Consequently, more additional damage occurs according to the bathtub model for the medium IPS as this model reports significantly more interior damage for buildings.

### 3.3 Flood damage for future scenarios

The developed flood risk assessment framework was applied to a set of sea level rise scenarios for the reference year of 2060. Flood damage was computed and used as a proxy for how flood damages and risk could evolve in future conditions. The set of seven scenarios is compiled in Tab. 1. As shown in Fig. 9a, a fully closed MOSE barrier keeps the peak flood level significantly below the safety threshold of 1.10 m ZMPS for the given meteorological event for all scenarios. A partially closed barrier would lead to a reduction of the flood peak by about 0.3 m for SLR0 and SLR1. Still, an open Lido inlet leads to high water

levels at Punta della Salute. Results suggest that the dampening effect by a partially closed barrier diminishes for SLR2. For a sea level rise of 0.45 m, the peak at the Piattaforma CNR would be at 2.25 m ZMPS, and the peak at Punta della Salute at 2.10 m ZMPS, implying that the damping effect is reduced by half.

It is noteworthy that for the 'allclosed' scenarios, SLR2 results in a slightly lower flood peak estimate than the other two

scenarios. A possible explanation is that for SLR2 the closure of the MOSE barrier occurs about 24 hours earlier relative to the flood peak, while for SLR0 and SLR1 it is closed about 4 hours before the flood peak. As the barrier is closed during a flood,

the part of the tidal wave that propagated into the lagoon before the full closure has more time to evenly spread out across the lagoon, resulting in a slightly lower average flood depth in the centre of the lagoon than for the other two scenarios. This ultimately influences the wind effect and maximum water levels at Punta della Salute.


Analysis of the implications of the different scenarios on the average inundation depths concludes that a partially-functioning MOSE barrier would significantly reduce the expected average flood depth for 90% of the buildings for sea level rise scenarios of SLR0 and SLR1. In SLR2, the increased sea level dominates over the dampening effect of the partial closure as visualized in Fig. 9b. This analysis also shows that for the storm surge of 12 November 2019, 50 % of all structures in Venice experienced

a flood depth of 0.55 m or higher. Only 10% of buildings experienced flood depths lower than 0.10 m and only 5% of buildings were not exposed to floods at all.

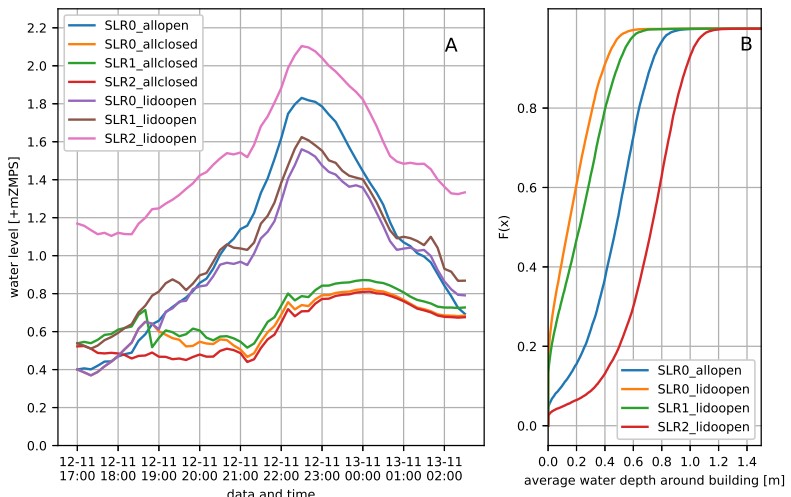

**Figure 9.** Flood depths for scenarios. a: Modelled flood peaks at Punta della Salute. MOSE barrier activation for the different scenarios was 12/11/19 18:40 (SLR0), 12/11/19 18:10 (SLR1) or 11/11/19 18:10 (SLR2) according to Tab. 4. b: Share of buildings exposed to certain average flood depths

Corresponding damage estimates for the different scenarios were computed using the calibrated INSYDE model. For the scenarios accounting for an (assumed) protecting MOSE barrier, the forecasting water level relevant to determine the height

of mobile protections at doors and windows was set to the safety threshold of 1.10 m ZMPS. As a result, the damage cost difference between medium IPS and risk averse IPS decreases with increasing flood depths. At the same time, the difference for the risk averse IPS is less apparent given that for SLR0-allopen, damages only occurred at the external walls, but for SLR0-

**Table 10.** Flood peak level at Punta della Salute [m ZMPS] and damage estimates [EUR million] for different scenarios

| scenario | peak level | d3dfm | | | bathtub | | |
|---|---|---|---|---|---|---|---|
| | | risk averse IPS | medium IPS | risk taking IPS | risk averse IPS | medium IPS | risk taking IPS |
| SLR0-allopen | 1.89 | 52.2 | 166.3 | 253.6 | 53.8 | 193.1 | 269.9 |
| SLR0-lidoopen | 1.56 | 37.1 | 95.0 | 132.0 | 39.7 | 119.7 | 156.9 |
| SLR0-allclosed | 0.82 | 0.0 | 0.0 | 0.1 | 0.0 | 0.0 | 0.1 |
| SLR1-lidoopen | 1.62 | 42.6 | 129.4 | 166.7 | 46.8 | 165.3 | 201.1 |
| SLR1-allclosed | 0.87 | 0.0 | 0.0 | 0.2 | 0.0 | 0.0 | 0.2 |
| SLR2-lidoopen | 2.10 | 179.7 | 289.6 | 309.4 | 196.3 | 300.8 | 320.0 |
| SLR2-allclosed | 0.81 | 0.0 | 0.0 | 0.1 | 0.0 | 0.0 | 0.1 |

lidoopen also partly on the inside due to lower protection levels. Results are compiled in Tab. 10.

An interesting observation can be made when comparing the damage estimates of SLR0-allopen to those of SLR2-lidoopen. Despite an approximately 0.21 m higher flood depth for SLR2-lidoopen, the effect on damage estimates for risk taking IPS and medium IPS are smaller than expected even though protection heights are on average also 0.40 m lower than in SLR0-allopen. Analysis of the formulations for vulnerability and exposure implemented in INSYDE provide a possible explanation: it is insufficient to replace the external and internal plaster that came in direct contact with the water. An additional height of one

meter must be replaced as well. Given that cost for plaster removal is independent of the required removal height, this implies that for a small flood depth, higher replacement costs occur already and are only incremented linearly for higher flood depths. As extreme flood depths are frequently lower than one meter, the influence of the additional height carries a stronger weight compared to the difference for higher water level scenarios.

**4   Discussion**

Venice is a city with a long history of flooding that is likely to extend into future despite the presence of the MOSE barrier. Until now, limited methodological approaches exist which provide estimations of future flood risk to structures and particularly to cultural heritage. This study developed a flood risk assessment framework that can be used for assessment of direct, tangible damages to residential and economic buildings, and can be extended in future research to account for the special conditions of

cultural heritage as well. The framework performs well compared to available damage claim data and gives some indications about possible future flood risk for extreme storm surges under a partially failing MOSE barrier system.

The developed hydrodynamic model provides reliable estimates of hazard characteristics inside the old-town. First, the validated hydrodynamic coastal model reproduces the flood peaks with an accuracy of $\pm 5cm$ despite some simplifications of the lagoon system, such as applying uniform meteorological conditions over the entire domain and neglecting freshwater inputs and wave action. Second, the cross-model comparison suggests that the hydrodynamic model performs as expected and may provide optimistic flood depth estimates inside the city as compared to the presently used static model (Liu et al., 2018). A final confirmation of flood depths inside the city by means of calibration and validation with flood depth records was not possible but should be a key focus in future studies as flood-enhancing components such as the sewage system, water coming from the ground, or wave influence were neglected. Those elements were not considered as no data on the 1D-network of the sewage systems and the other processes were available in due time and resources to investigate these data in field trips were not available. In addition, following from the comparison of parent and nested model depth estimates, a grid convergence analysis should be conducted to find the optimal grid resolution for the city of Venice. Despite a grid size of 1.3m near structures, which is already rather high compared with other hydrodynamic urban models (Xing et al., 2019), the specific setting of Venice with its narrow street system may require increasing the resolution even further.

Some modelling challenges of the hydrodynamic model have to be highlighted. Due to the complex urban structures and altimetry, some extreme local water levels that occurred in the parent and nested models were likely caused by the complex grid structure and the algorithm describing the wetting and drying process inside the model (Deltares, 2021). This led not only to incorrectly high flood depths at a few buildings but also prevented the consideration of one of the nested sub-models. Part of the instabilities can be solved by grid refinement, bathymetry alteration, or adjusting the modelled time periods. In accordance with previous studies (Scorzini and Frank, 2017; Arrighi et al., 2013), it was found acceptable to use bathtub flood depth estimates for the remaining structures instead, given the limited influence of flood depth variation on the damage estimate. However, while the current set-up of the hydrodynamic model results in roughly similar damage estimates as the bathtub model, a fully functioning hydrodynamic model may add additional benefits to the flood risk assessment framework as it can account for (changing) physical characteristics explicitly, allow for a calibration based on flood depth information, and incorporate additional flow path-components such as a 1D sewage system, which might lead to different flooding patterns.

The adjusted version of the INSYDE damage model is able to reproduce the total damage claim volume related to the storm event of 12 November 2019 as shown in Tab. 8. Analysis of the sub-set of immediate response damage claims also confirm initial expectations of relatively high individual protections levels in Venice as frequent and intense experience of flooding have been reported to contribute to higher levels of individual flood preparedness (Kreibich et al., 2015). Moreover, results imply that the effect of protection measures has a strong influence on the estimated damages. It is important to note that damages were only caused by the inundation depths and not by flow velocities or flood duration according to the INSYDE model. Flow velocities inside Venice and near its buildings were lower than the required threshold (0.5 m/s) for more than 95 % of the buildings, as shown in Figure S13 in the supplementary material. Similarly, inundation duration had no damage-mediating

effect because it did not exceed the pre-defined threshold of eight hours for the analysed flood events as shown in Fig. 9.

However, the poor structure-wise depth-damage correlation and the alignment of the two considered sets of reported damage claims with different (combinations) of IPS reiterate commonly faced challenges of flood damage modelling (Ahn et al., 2019; Diaz Loaiza et al., 2022). Limited knowledge of the system introduces uncertainty in the damage estimates. As an example, about half of all damage claims ( 7,644) were linked to about 20% of the structures in Venice only. Meanwhile, 90% of structures were found to be exposed to an average flood depth of at least 0.1 m according to the hydrodynamic model. Thus, it is questionable whether exposure and vulnerability of the system are adequately represented given that modelled damages of external walls alone are almost as high as the reported damages. In addition, preparedness was simplified as perfectly functioning mobile barrier systems installed at all buildings, like in this study. However, protection levels have been reported to be very diverse and could also (partially) fail to provide the promised level of protection in reality. Additionally, more protection measures may be in place to reduce the flood damages. Moreover, many exposure and vulnerability relations of the synthetic damage model were transferred unaltered, despite the possibility that they may not reproduce the present hazard-structure interaction processes in Venice.

At the same time, limitations of the available damage claim data-sets have to be accounted for as well. It can generally be questioned whether reported damages represent the full set of effective damages of a flood event. Potential claimants may have opted to undergo significant bureaucratic efforts for (sometimes) limited financial support (Molinari et al., 2020). Alternatively, claimants may not have seen the need to replace (some) damaged elements, e.g. because of their experience with frequent flooding. Marks of previous floods at house fronts throughout the old-town support this hypothesis. Additionally, given that the available damage data are spatially and/or component-wise aggregated, limited conclusions can be drawn from the damage data analysis to address the mentioned limitations of the framework. While the structure-wise analysis of immediate response claims allowed for a comparison of the bathtub model and hydrodynamic model, the high-aggregation level of all damage claims in combination with the numerical challenges in the hydrodynamic sub-models, did not allow to confirm findings of this comparison. Information from a detailed investigation of the effective and reported damages for the 12 November 2019 flood event may provide required additional confidence in the developed damage model. Also, a thorough analysis of the variety and spatial distribution of building types and installed preparation and protection measures on structure and neighborhood level, as well as other vulnerability characteristics, in Venice would be required for a better representation of the system.

When discussing the accuracy and reliability of the applied damage model, it is also worth considering that another study analyzing exceptionally extreme flood events suggests much higher flood damages (Caporin and Fontini, 2014); for flood events exceeding 1.80 m ZMPS, damage estimates amount to EUR 196.33 million[18] even though only the refurbishment (plastering) of walls is considered. Given the varying approaches, many reasons could contribute to the diverging damage estimates. Two striking reasons were identified: estimates of the buildings requiring special care due to their historical importance diverge for

---

[18]Price level of 2013, not adjusted for inflation.

the two studies (present study: 25% of buildings declared as cultural heritage, in other study 50% of buildings) along with the corresponding increase in refurbishment cost (present study: 10%, other study 50%). It is also important to acknowledge that considering an economic value of cultural (world) heritage in terms of increased re-construction costs does not holistically represent the flood impact on a cultural heritage sites and assets. Firstly, impact on the cultural value is not represented in terms of reconstruction costs. Secondly, it is unknown to what extent cultural heritage value can be restored or reconstructed after being damaged or destroyed. Both aspects are not addressed in the current set-up of the damage model. Transparent and robust cultural heritage decision making should include a wide range of heritage values while recognizing that these can change over time and should be regularly updated (Fatorić and Seekamp, 2018). Additionally, the assumed basis reconstruction costs may vary: in the present study, reconstruction cost values from another region were used under the assumption of limited variation across Italy. Further investigation into possible differences and use reconstruction cost information for the Veneto region is recommended instead[19].

Results on the effect of the MOSE barrier on the water level inside the lagoon align with previous studies, suggesting that a partial closure will still cause flooding of the old-town of Venice (Umgiesser et al., 2021). The study adds to the existing knowledge as it considers the second most extreme flood event experienced, while previous studies have mainly investigated more frequent, less extreme flood events (Zampato et al., 2016; Vergano and Nunes, 2007). The present study adds new insights suggesting that the damping effect of a partially closed MOSE barrier on the flood wave will reduce as sea level rises and may consequently amplify flood risk in future. To confirm this finding in future studies, some of the present's study limitations should be addressed: for the applied future scenarios, present conditions of the system were used. However, the sediment budget of the lagoon is negative, meaning that the lagoon is currently deepening and may look significantly different in 40 years from now (Tambroni and Seminara, 2006). The same applies for local subsidence processes which have significantly contributed to flood risk in the past and may continue to do so in future as well (Zanchettin et al., 2021). Also, variation in tidal amplitude due to changes in bathymetry and mean sea level as observed in the past, may continue in future as well (Ferrarin et al., 2015).

In addition, some inaccuracy regarding the flood levels is likely to be introduced as processes of seepage through the barrier and freshwater input in the lagoon have been neglected in the present study. This is particularly relevant for SLR2, where the MOSE barrier would be closed for more than 36 hours. In previous studies it has been suggested that seepage through the fully closed barrier could result in water level increase between 0.27 cm to 2.1 cm per hour (Umgiesser and Matticchio, 2006). Consequently, peak water level could be expected to be about 8.1 to 63 cm higher for SLR2-allclosed, while the effect of seepage could add between 1 and 8.4 cm in a SLR0-allclosed scenario where MOSE closure happens about 4 hours before the flood peak. Seepage and freshwater input may also increase water levels for scenarios with an open inlet at Lido.

---

[19]accessible here: https://www.regione.veneto.it/web/lavori-pubblici/prezzario-regionale

The results of the scenario analysis highlight the importance of a fully functioning MOSE barrier and the damage mediating influence of the individual protection scenarios. In line with previous studies investigating the remaining flood risk under climate change with a fully functioning barrier (Nunes et al., 2005), the present study suggests that a fully closed MOSE barrier limits the effect of flooding for the considered meteorological flood event to very few buildings inside the old-town with very small damages for all considered sea level rise scenarios as shown in Tab. 10.

Even though the applied methodology to represent preparedness and individual flood risk protection by means of different IPS and their effectiveness has mainly a conceptual value, some insights can nevertheless be derived: the warning level and how residents will respond to this in terms of individual protection in light of a (expected) functioning MOSE barrier appear to have significant influence on the expected damages as shown in Tab. 11. Table 11 gives the change of estimated damage for the different scenarios relative to the modelled damages for the flood event of 12 November 2019 represented by SLR0-allopen. It shows that a partially functioning MOSE barrier could reduce damages of a storm surge event like that on 12 November 2019 by 17% to 48% for SLR0 or SLR1 under the assumption of unaltered levels of individual protection in future. The reduction is strongest for the SLR0-lidoopen scenario, assuming a (constant) risk-taking IPS, where damage would be reduced to 52% of the estimation for SLR0-allopen. As discussed, the damping effect of a partially closed barrier diminishes for SLR2-lidoopen. As a result, damages could increase by a factor 1.08 to 3.44 if sea level rise follows the pessimistic prognosis of climate change.

At the same time, individual protection levels may change in future depending on the performance and reliability of the MOSE barrier. In the worst case, meaning that protection levels change from a risk averse IPS to a risk taking IPS, damages could be up to 5.92 times higher compared to flood damages of SLR0-allopen as shown in Tab. 11. Compared with a scenario where the individual protection level remains constant, damages would be about 72% higher in this case. At the same time, in the case that individual protection levels increase from an medium IPS to a risk averse IPS, damages could be reduced to 26% for SLR1-lidoopen or just slightly increase by 8% in case of SLR2-lidoopen.

**Table 11.** Ratio of future flood damages and SLR0-allopen under varying IPS (developments in future). I: risk averse IPS, II: medium IPS, III: risk taking IPS.

|  |  | SLR0_lidoopen | | | SLR1_lidoopen | | | SLR2_lidoopen | | |
|---|---|---|---|---|---|---|---|---|---|---|
|  |  | I | II | III | I | II | III | I | II | III |
| SLR0 allopen | I | 0.71 | 1.82 | 2.53 | 0.82 | 2.47 | 3.19 | 3.44 | 5.54 | 5.92 |
|  | II | 0.22 | 0.57 | 0.79 | 0.26 | 0.78 | 1.00 | 1.08 | 1.74 | 1.86 |
|  | III | 0.15 | 0.37 | 0.52 | 0.17 | 0.51 | 0.66 | 0.71 | 1.14 | 1.22 |

As present knowledge of influencing drivers of future flood risk is very limited, this study is only a starting point for a more concise analysis of the implications of the MOSE barrier on the old-town of Venice and the individual protection levels in particular. At this point, it is unknown what effect the operational MOSE barrier will have on the early-warning system in

Venice and the level (and types) of installed protection measures by residents. Additionally, the provided estimates are all based on present monetary values and present exposure and preparedness conditions. They are expected to change in future, again depending on both possible socio-economic and political developments and the reliability of the MOSE barrier to protect the old-town and its residents in the future.


## 5   Conclusions

In this study, a flood risk assessment framework has been developed, proving able to reproduce the flood event of 12 November 2019 with an accuracy of $\pm 5 cm$ in the proximity of the old-town and providing damage estimates in accordance with available damage claim data. While the use of a hydrodynamic model posed some numerical challenges and resulted in similar flood
damage estimates than based on a bathtub model, the opportunity to integrate additional elements such as wave-effects or a 1D-flow path-component representing the sewage system in the low-lying city might allow for a more accurate flood hazard estimation beneficial for efficient flood risk management. The implemented damage model can reproduce damage claim data but faces commonly acknowledged uncertainties due to limited knowledge about the system and damage processes. Various existing approaches and elements (hydraulics, damage model, interventions, sea level rise scenarios) were integrated to develop
and test a novel approach to risk assessment for Venice. While the application focus of this study focuses on events with a single return period the framework can be easily used to consider other events (with other return periods) to come to a complete risk assessment in current and future conditions and for various interventions. Given the complexity of the system and the large numbers of possible interventions, it would be a study by itself to evaluate all the (combinations) of interventions (Berchum et al., 2019). Thus, in this paper we have focussed on the introduction of the framework and its illustration for a limited number
of events and interventions.

Developing a methodical risk assessment framework for the cultural heritage city has provided some valuable insights into expected flood exposure and damages in the old-town of Venice. While this study confirms the general appropriateness of the MOSE barrier to protect the city of Venice for extreme storm events for additional rising sea level up to 45 cm, it was also
found that the damages in case of a partially closed MOSE barrier may still increase significantly for most considered scenarios. While an improved individual protection level in future could lead to a damage reduction of up to 78% for present sea level and 74% for an optimistic sea level rise prognosis, damages could be up to 1.08 to 5.92 times higher in 2060 in case of unchanged or decreased level of individual protection. Based on the findings of relative importance of individual flood protection in light of a potentially failing MOSE barrier, this study provides indication that a better understanding of presently applied
flood protection is needed to identify realistic individual protection scenarios for future conditions. This would be helpful to identify possible areas of action to maintain (or advance) existing structure-wise flood protections and individual preparedness. In addition, the influence of the MOSE barrier on the reported warning levels and the effectively installed protections was identified as an important question to address in order to reduce flood risk in Venice until 2060. As such, the proposed flood

risk assessment framework provides a methodical approach useful to support future decisions on flood risk management.


Additional studies should be done to improve the presented framework. Addressing some of the limitations, particularly the simplification of the system by excluding the sewage system, grid instabilities and lack of calibration data, may add additional confidence to the exposure modelling. Moreover, incorporating information on future return levels of storm events as well as failure probabilities of the MOSE barrier should be addressed and incorporated in the present framework to allow for a proper

flood risk assessment to support the efficient and effective allocation of (additional) resources to flood protection in Venice. Also, a better understanding of the spatial distribution of protection measures and other damage mediating characteristics within the districts of the old-town, ideally for each structure, is required for a better representation of the system. Additionally, new building types in the damage model can be implemented to account for some characteristic cultural heritage buildings as proposed in the supplementary material. This would contribute to a better and multidimensional understanding of the present

and future flood risk.

*Code and data availability.* Files and data used for the hydrodynamic and damage modelling are made available on the following repository along with an explanatory overview document: https://1drv.ms/u/s!AujDMT3F11JwgpEoTj2zfvrJqDcOdA?e=dY2c6O

*Author contributions.* The paper is product of the M.Sc. thesis work of JS. JS was responsible for the progression of research, the model

runs and the post-processing analysis and writing the paper. CF provided data and information regarding the 12 November 2019 flood event and contributed to the analysis of the hydrodynamic and flood modelling results. BJ was chair of the M.Sc thesis, reviewed the paper and contributed to defining the general scope and approach of the study. ADL provided support on the hydrodynamic modelling and writing process. AA supported the communication with Italian official entities. SF provided data and information on cultural heritage evaluation. CF, BJ, ADL, AA and SF also contributed with discussion and revision.

*Competing interests.* The authors declare that they have no conflict of interest.

*Financial support.* Christian Ferrarin has been supported in this work by the STREAM project (strategic development of flood management, project ID 10249186) funded by the European Union under the V-A Interreg Italy-Croatia CBC program.

*Acknowledgements.* We would like to issue special thanks to Dr. A. R. Scorzini for her immediate support and for sharing her insights and experience in the complex field of damage modelling in the context of Italy. Furthermore we would like to thank the office of the delegated

Commissioner Delegate for the Emergency resulting from the exceptional tide of 12 November 2019 in Venice for their willingness and cooperation in providing statistical data related to the declared damages and in particular M. Calligaro for his valuable and extensive effort to provide all possible damage claim information. Finally, we want to thank G. M. Lemos for sharing insights and data from her experience in D3DFM modelling of the Venetian lagoon.

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
