# Peer review of "Developing a framework for the assessment of current and future flood risk in Venice, Italy"

_Natural Hazards and Earth System Sciences, 2021_

## Author Comment (AC1)

Dear Reviewer, first we want to thank you for your constructive, in-depth and clear questions and comments. Below you can find our answers point-by-point to your comments. We also highlight (where possible) the part of the original submitted manuscript that have been modified to address your comments. To facilitate the reading, we added our responses to your comments in red. If changes to the text are proposed, changes are underlined.

**Major Reviews:**

The author are using an interesting and complex combination of hydrodynamic models to simulate the flood hazard in Venice; yet, the authors are also only using water depth as the descriptor of flood damages in the case study area. One of the main advantages of hydrodynamic models is the capability of providing information pertaining to the duration and velocity of a flood event, which however are not considered in the damage modelling framework. As such, the value-added of utilising the nested hydrodynamic models is not clear in the current version of the manuscript, and seems to only be adding to the complexity of the proposed methodological framework. While the authors are already using a hydrodynamic model to simulate the dynamics from the Adriatic sea to the Lagoon of Venice (i.e. parent model), I would strongly suggest the authors to highlight the benefits of using the nested hydrodynamic models over a more simplistic bathtub approach for the studied application.

Thank you very much for this question. Based on your comment, we clarified our motivation for applying a 2D-hydrodynamic model in line 135ff :

*"Studies looking into the distribution of flood depths in Venice have used a static model, also called bathtub model (Cellerino et al., 1998). It uses the water level at the tidal gauge of Punta della Salute and compares it with the surface elevation of the old-town of Venice to identify the flood extent and depth. A bathtub model assumes instantaneous flooding. It neglects the process of flood wave progression and therefore could overestimate the flood depths inside the city. Using a 2D hydrodynamic model might be able to capture the flood progression into the city, the role of sewage networks and other processes more realistically while also providing the appropriate framework to account for other flood parameters such as flow velocity. Moreover, the hydrodynamic model can be forced with variable water levels at the boundaries of the nested sub-models, thus accounting for strong water level gradients over the city registered by the observations during the 12 November 2019 event."*

To address the relevant comment about the flood velocity as a damage descriptor, we added the following in line 414: "*It is important to note that damages were only caused by the inundation depths and not by flow velocities or flood duration according to the INSYDE model. Flow velocities inside Venice and near its buildings were lower than the required threshold (0.5 m/s) for more than 95% of the buildings, as shown in Figure S13 in the supplementary material. Similarly, inundation duration had no damage-mediating effect because it did not exceed the pre-defined threshold of eight hours for the analysed flood events as shown in Fig. 9.*

In the same topic of the last point, it would be interesting to provide a table similar to Table 7 but comparing the results of the hydrostatic and the hydrodynamic models, in terms of R and RMSE.

Thank you for this comment. Table 7 only refers to the parent model (covering the Venetian lagoon). The stations mentioned in the table are tidal gauge stations spread over the lagoon. However, we did not apply the bathtub model for the entire lagoon but only inside the city of Venice. A

comparison of flood depth-estimates derived from the bathtub model and from the nested hydrodynamic model is shown in Figure 6a.

As Figure 6a shows an underestimation flood depth bias by the d3dfm model with respect to the bathtub approach, It would be interesting to include also the damage results from the bathtub flood model, if possible, so to provide a meta-model comparison. The same rational is valid for Tables 9 and 10.

We thank the reviewer for this suggestion. We added the information to the respective tables and also added some additional textual analysis along with them.

**Table 8.** Comparison of damage claims and estimates based on hydrodynamic (d3dfm) and bathtub (btb) flood depth estimates [EUR million]

| | | | INSYDE | | claims |
| | | | d3dfm | btb | |
|---|---|---|---|---|---|
| sub-set of structures | | risk averse IPS | 12.9 | 13.1 | |
| | | medium IPS | 42.0 | 47.5 | 25.7 |
| | | risk taking IPS | 63.1 | 65.8 | |
| all structures | | risk averse IPS | 52.3 | 53.8 | |
| | | medium IPS | 166.3 | 193.1 | 56.2 |
| | | risk taking IPS | 253.6 | 269.9 | |

New text added in line 312: *"It can also be seen that in line with the lower level of flood depth estimations by the hydrodynamic model, damage estimates based on the bathtub calculations are generally larger. The difference increases with decreasing level of individual protection."*

**Table 9.** Performance indicators of damage estimates based on hydrodynamic (d3dfm) and bathtub (btb) flood depth estimates for structures with immediate response claims

| | | risk averse IPS | expected IPS | risk taking IPS |
|---|---|---|---|---|
| d3dfm | R [-] | 0.22 | 0.26 | 0.26 |
| | RMSE [EUR] | 19,382 | 22,158 | 29,332 |
| | RE [%] | 308.9 | 87.8 | 55.5 |
| btb | R [-] | 0.22 | 0.25 | 0.26 |
| | RMSE [EUR] | 19,384 | 23,298 | 30,122 |
| | RE [%] | 304.9 | 71.5 | 51.8 |

New text added in line 317: *"Both indicators suggest that the damage claims might be slightly better estimated for damages computed based on bathtub flood estimates. Furthermore, claims might be slightly better estimated based on an expected IPS or risk taking IPS for most buildings."*

**Table 10.** Flood peak level at Punta della Salute [m ZMPS] and damage estimates [EUR million] for different scenarios

| scenario | peak level | d3dfm | | | bathtub | | |
|---|---|---|---|---|---|---|---|
| | | risk averse IPS | expected IPS | risk taking IPS | risk averse IPS | expected IPS | risk taking IPS |
| SLR0-allopen | 1.89 | 52.2 | 166.3 | 253.6 | 53.8 | 193.1 | 269.9 |
| SLR0-lidoopen | 1.56 | 37.1 | 95.0 | 132.0 | 39.7 | 119.7 | 156.9 |
| SLR0-allclosed | 0.82 | 0.0 | 0.0 | 0.1 | 0.0 | 0.0 | 0.1 |
| SLR1-lidoopen | 1.62 | 42.6 | 129.4 | 166.7 | 46.8 | 165.3 | 201.1 |
| SLR1-allclosed | 0.87 | 0.0 | 0.0 | 0.2 | 0.0 | 0.0 | 0.2 |
| SLR2-lidoopen | 2.10 | 179.7 | 289.6 | 309.4 | 196.3 | 300.8 | 320.0 |
| SLR2-allclosed | 0.81 | 0.0 | 0.0 | 0.1 | 0.0 | 0.0 | 0.1 |

During recent MOSE activations, the Malamocco inlet was left open, while both Lido and Chioggia were closed, as this is the main inlet for commercial and industrial ships. While the scenario "lidoopen" is certainly very interesting and capable of providing much appreciated information to decision-makers and to the general public, it would be also interesting to consider, if possible, a more plausible and realistic scenario were the Malamocco inlet only is left open.

We thank the reviewer for this suggestion. We indeed considered to investigate a larger set of alternative closing scenarios, but due to time constraints we limited ourselves to two closure states. With the paper we wanted to focus on possible, critical scenarios (unintentional non-closure of one of the barrier gates) rather than most likely scenarios (open closure for shipping purposes). In a preliminary a-priori assessment we identified the Lido inlet as the possible critical one given its proximity to the old-town and the larger cross-section compared to the other two inlets. This choice is in line with previous works using the same line of argumentation (Umgiesser, 2020) or providing evidence that partial closure of the MOSE barrier (closing Lido & Chioggia inlet) can have a relevant water level managing effect (Cavallaro et al., 2017)

To be more explicit on the motivation of choice, we added a sentence in line 123: "*Regarding the MOSE barrier, two closure states are considered: a fully functioning MOSE barrier ('allclosed') and a set-up where all inlets but the Lido inlet close ('lidoopen'). Previous works (Mooyaart & Jonkman, 2017; Vrancken et al., 2008), and experiences from practice in Venice (Colamussi, 1992; Umgiesser and Matticcio, 2006) have shown that there is a probability of (unintentional) non-closure of storm surge barriers resulting in possible, critical scenarios. In an a-priori assessment of the inlets with regards to their dimensions and proximity to the old-town of Venice, we identified that non-closure of the Lido inlet ('lidoopen') is likely the most critical partial-closure scenario. This choice in line with previous studies indicating the prominent importance of this inlet to manage water levels in Venice (Cavallaro et al., 2017; Umgiesser, 2020).*"

The authors of this work agree that this framework could be further used to analyse alternative closure scenarios.

In line 142, the authors mention that the d3dfm model "allows to account for additional processes like wave action or 1D flow of the sewage system". From the manuscript's text, these are not taken into consideration. In the specific case of Venice, it could be relevant to consider the effects of the sewage and drainage system when modelling high tide floods, as during high tide events water may come directly from underneath the city instead of overflowing from the canals. The non-consideration of such phenomena might lead to the underestimation of flooded areas, especially in a scenario such as Venice, where buildings are often attached to one another, leading to significant areas isolated from overland flow in the perspective of the hydrodynamic model. Indeed, this might

be the case why Figure 6a shows that the hydrodynamic model underestimates flooding in the majority of cells with respect to the hydrostatic (bathtub) approach. Could the authors better explain why this option has not been included in the flood modelling framework?

We thank the reviewer for their relevant question. Indeed, we initially planned to account for alternative processes (including the sewage system). For clarification we added the following additional sentence starting in line 394: "*Those elements were not considered as no data on the 1D-network of the sewage systems and the other processes were available in due time and resources to investigate these data in field trips were not available.*"

It is not clear from the text if the seven nested sub-models exchange information among themselves as boundary conditions or just with the parent model? Please better explain the nested setup.

We thank the reviewer for spotting this unprecise formulation. We have modified the sentence in line 149 as follows: *"Water level time-series from the parent model simulation were extracted at 168 locations inside and around the old-town of Venice. Every nested model is enclosed by a sub-set of these locations. Consequently, for every nested model, the water level time-series of the enclosing locations were used as the boundary inputs driving the hydrodynamic simulation. As such, the sub-models did not exchange information among each other but were run independently."*

Regarding the altimetry data that has been used in this study, has the correction to the ZMPS datum been done within the work developed in this manuscript or is it a data that has been obtained as already published from other sources? If the correction to the reference ZMPS level has been done as part of the work developed in this manuscript, please add a methodological description on how this has been performed (I suggest adding it in the supplementary material if possible).

The correction to ZMPS datum has been done within this work. We added a new sub-section "Conversion of altimetry data" and add an explanatory schematisation:

*"Altimetry data were derived from various sources with varying reference datum as shown in Tab.2 of the main paper. Conversion of the altimetry to the ZMPS datum were performed by adding or subtracting the absolute difference between the respective datum and ZMPS as conceptualised in Fig. X. Accordingly, the bathymetry information of the Venetian lagoon and the canals within the old-town (both available in IGM42, where 0m IGM42 corresponds to + 0.23 m ZMP ) were corrected by subtracting 0.23cm from the original IGM42-referenced altimetry data. The surface of the old-town of Venice (also provided in IGM42) was corrected by adding 0.23 cm given that the surface was generally located above the respective MSL. Altimetry information of the Adriatic shelf were provided with reference to the LAT datum which was assumed in this work to approximately correspond to -0.4m ZMPS. Consequently, bathymetry of the Adriatic shelf was corrected by adding +0.4 m."*

Altimetry old-town Venice

IGM42 datum

0.23 m        ZMPS datum

0.40 m        LAT

Bottom profile Venetian lagoon

The authors refer to "grid instabilities" in lines 298 and 299. Could the authors better explain what are those instabilities and how are they defined? Also, it is not clear if the whole Castello sub-model was affected or just part of it (14% of total, where total refers to the Castello sub-model or to Venice?). Please better explain.

We thank the reviewer for this in-depth question. When referring to "grid instabilities", we mean that during the model run to compute water level time-series in all grid elements, water levels start to increase locally to extreme levels. In the 'Castello' sub-model these instabilities were locally extreme (in the order of 10-20 meters (ZMPS)). At other locations, they only had a minor effect. According to the D3DFM-documentation, such instabilities can occur at dry cells that contain only a very limited amount of water. Also, from modelling experience it has been found that instabilities could occur if grid element resolution is varying strongly at a certain location, or altimetry information is changing too drastically.

Although many attempts were made to solve these instabilities (going even so far to re-create the entire sub-model 'Castello'), no reason/solution could be found. Finally, we decided to ignore the hydrodynamic results for sub-model 'Castello' and use the bathtub information instead. Consequently, for all 2,098 buildings located in sub-model 'Castello' (about 14% of all buildings in the old-town of Venice) flood depth estimates could only be derived using the bathtub model.

**Minor Reviews:**

Some of the graphics are difficult to read due to their low quality and/or small font size (e.g. Figure 9).

Thank you for highlighting this, we have updated all the graphics to .eps and partially increased font sizes.

I might be wrong, but Figure 8 may be out of scale on the y-axis for the variable F2 (risk taking IPS).

Thank you very much for spotting this. Indeed, the figure was out of scale. The figure has been updated.

Figure 9a, as a suggestion to improve the readability of the figure, it would be interesting to add some indications on when MOSE is activated and deactivated.

Thank you for this comment. Since the figure is already rather complex, we decided to add an elaboration in the caption text instead. It was changed to: *"Flood depths for scenarios. a: Modelled flood peaks at Punta della Salute. MOSE barrier activation for the different scenarios was 12/11/19 18:40 (SLR0), 12/11/19 18:10 (SLR1) or 11/11/19 18:10 (SLR2) according to Tab. 4."*

Line 2-3, sentence "limited scientific knowledge of flood hazard and flood damage modelling of the old-town of Venice is available to support decisions to mitigate existing and future flood risk."; I would suggest to rephrase the sentence, as flood hazard information is available, including publicly-available flood maps and walking paths covering the historical city centre of Venice for different flood quotas. Instead, information about flood risk is definitely much less available.

Thank you for this comment. We rephrased line 2-3 to: *"Despite this existence-defining condition, limited scientific knowledge on flood risk of the old-town of Venice is available to support decisions to mitigate existing and future flood impacts."*

Some sentences are unclear and/or could be better structured (e.g. line 1, "Flooding has been a serious struggle to the old-town of Venice, its residents and cultural heritage and continues to be a challenge in the future."). Also, spelling is mixed between British and American styles (e.h. behaviour vs. behavior; analyse vs. analyze). An in-depth proof-reading of the manuscript is recommended.

Thank you for this comment. We reviewed the manuscript thoroughly and corrected spelling errors. We changed the sentence in line 1 to: *"Flooding causes serious impacts to the old-town of Venice, its residents and cultural heritage."*

Line 2 and line 51; is the term "existence-defining" correctly employed (particularly in the phrase of line 2)? Or should be "existence-defying"?

Thank you for this comment. We think that both terms are usable. The situation is existence-defining as a lot of daily life is affected (and partly limited) by flood risk. But it could also be interpreted as existence-defying because of the struggle caused by flooding that makes the old-town rather inhabitable. From our perspective, the narrative along the lines of existence-defining makes more sense, since people still choose to live there, even though limited knowledge is available on the (development) of flood risk.

Line 24; please correct the definition of exposure as, following the IPCC definition, it is not only related to human systems, but to the "inventory of elements in an area in which hazard events may occur". The next phrase on the manuscript highlights this, and the text should be consistent (i.e. "human health, environment, cultural heritage and economic activities").

Thank you for this comment. We reviewed the manuscript thoroughly and corrected spelling errors. The definition of exposure terminology has been adjusted in line 24 as follows: "*According to the IPCC, flood risk is defined as the combination of a specific hazardous flood event, elements (i.e. infrastructure, people, livelihoods, environment, and cultural, social and economic assets) which might be exposed to a hazard in a certain area, and the vulnerability of these elements, meaning predisposition to be adversely affected \citep{Field.2012}."*

Line49: The phrase "Additionally, intangible damages to cultural heritage sites and their meaning for the cultural identity of the region and nation can be expected (Wang, 2015)" is not very informative and could either be removed or extended with some examples of intangible damages to cultural heritage sites.

We extended the existing sentence in line 49 with a few examples which have been reported for the disastrous flood of 1966 in Florence as follows: *"Additionally, intangible damages to cultural heritage sites (e.g. loss of historic books or documents, damage to iconic paintings) and associated effects on the cultural identity of the region and nation can be expected \citep{Wang.2015, Arrighi.2018}."*

Please define all acronyms in the main text (e.g. ZMPS, IGM42, etc.).

We added the definition for the acronyms ZMPS, IGM42, LAT and RMSE.

As a suggestion, the paragraph between lines 69 and 75 could be moved to the methods section.

We thank the reviewer for their suggestion and agree with it. We therefore moved the paragraph between l.69 and l.75 along with figure 1 to the methods sections.

Please better explain the phrase in line 150 "All grid points inside a 4m buffer around each structure were used to derive an average water level". Does this means that water level per building considers only the surrounding flooded pixels or all pixels? Please clarify?

To clarify, we modified the sentence in line 150: *"Within each nested model, the maximum water level per building was derived by taking into account the maximum water levels of every grid point within a 4m distance from the building perimeter."*

Unless supported by a reference or data, I would suggest renaming the "Expected IPS" scenario to something else that better fits with the assumptions and discussions, such as "Risk neural IPS".

We want to thank the reviewer for this comment and totally agree. Expected IPS has been renamed to medium IPS throughout the manuscript (and figure legends). We did not use the term risk neutral as it is also used to define indifference about risk.

---

## Author Comment (AC2)

Dear Reviewer, first we want to thank you for your constructive, in-depth and clear questions and comments. Below you can find our answers point-by-point to your comments. We also highlight (where possible) the part of the original submitted manuscript that have been modified to address your comments. In order to facilitate the reading, we added our responses to your comments in red. If changes to the text are proposed, changes are underlined.

1. The authors claim that they propose a risk assessment framework for the city of Venice. However, the study describes a flood damage estimation, for a particular event, using standard methods that are commonly employed in such types of assessments (several examples of such studies are cited in the manuscript). In which way(s) is this framework or method that the authors propose novel and how does is this different from previously proposed frameworks (e.g. the one the IPCC employs)? This needs to be clearly described in the manuscript. My understanding from reading the manuscript is that the study describes a detailed assessment in terms of flood damage estimates for a specific event under a range of scenarios using established methods; this is an important study but does not involve any methodological innovations. If this is the case and I have not missed something, then this should be communicated accordingly in the manuscript.

We thank the reviewer for this critical comment. Indeed, this study uses one specific event to derive damage estimates that can be compared against available damage claim data to calibrate the used framework. To provide a clearer discussion of this aspect, we add the sentence in line 75 (the paragraph from line 69 to line 75 is moved to the methods section based on the comment of the other reviewer): "*This simplification was used as information about (future development of) return periods of the studied storm surge event, and probabilities of barrier failure scenarios are not available. However, the derived development of flood damage estimates as provided in this study could be easily translated into flood risk information by accounting for the probabilistic information.*"

The authors agree that the conceptual idea and applied methodology of this work are per se not novel. However, although several studies have focussed on flood risk models for delta cities around the world no such flood risk model is available for Venice as elaborated upon in in the manuscript line 60 ff. The present work also adds several novel aspects to flood risk assessment in Venice: detailed hydrodynamic and damage models (incl. cultural damage) and the inclusion of barrier failure scenarios.

2. I am a little concerned regarding the calculation of damages to cultural heritage – although the authors describe very clearly the method used for this calculation, actual damages of world heritage cannot always be substituted simply by higher expensive building costs (and I am not only talking about intangible damages). I'm worried that this calculation of tangible damages leaves a feeling that such damages are possible to address with increased investment, which is not the case. I am not suggesting that what the authors have done is not useful but would propose that they spend a few lines in the discussion to address this point.

Again, we thank the reviewer for this critical and thoughtful comment. We second this concern. Thus, we have developed (but unfortunately not applied) a more holistic conceptual idea about how to assess flood risk of cultural heritage which can be found in the supplementary material. To address the concern regarding the calculation of tangible damages to cultural heritage we added in line 456: "*It is also important to acknowledge that considering an economic value of cultural (world) heritage in terms of increased re-construction costs does not holistically represent the flood impact on a cultural heritage sites and assets. Firstly, impact on the cultural value is not represented in terms of reconstruction*

*costs. Secondly, it is questionable to what extent cultural heritage value can be restored or reconstructed after being damaged or destroyed. Both aspects are not addressed in the current set-up of the damage model. Transparent and robust cultural heritage decision making should include a wide range of heritage values while recognizing that these can change over time and should be regularly updated (Fatorić and Seekamp 2018)."*

3. The model setup, in particular the nesting (e.g. boundary conditions), is not very clearly described.

We thank the reviewer for this comment. To address it, we added the following adjustment in line 149: *"Water level time-series from the parent model simulation were extracted at 168 locations inside and around the old-town of Venice. Every nested model is enclosed by a sub-set of these locations, as shown in Fig. 3. Consequently, the water level time-series of the enclosing locations were used as the boundary inputs driving the hydrodynamic simulation for every nested model. As such, the sub-models did not exchange information among each other but were run independently."*

4. I find interesting that the authors suggest that the use of the bathtub model results in acceptable damage estimates, as in other studies. Would this mean that we could avoid the computational and time costs related to the application of the hydrodynamic model? Or to what extent would this be possible? Maybe an extra line or two discussing this would be useful (just a suggestion).

We thank the reviewer for this comment. Indeed, it might be a conclusion that could be drawn from the analysis. To make our point more clear, we adjusted line 411 ff: *"However, while the current set-up of the hydrodynamic model results in roughly similar damage estimates as the bathtub model, a fully functioning hydrodynamic model may add additional benefits to the flood risk assessment framework as it can account for (changing) physical characteristics explicitly, allow for a proper calibration, and incorporate additional flow path-components such as a 1D sewage system which might lead to different flooding patterns."*

Finally, I have listed below some further (secondary) points that would require clarification:

- Line 3: limited information of flood hazard? I would think that this is not the case in Venice?

Thank you for this comment. We rephrased line 2-3 to: *"Despite this existence-defining condition, limited scientific knowledge on flood risk of the old-town of Venice is available to support decisions to mitigate existing and future flood impacts."*

- What is the return period of the modelled event? This is an important parameter, particularly when assessing risk.

Thank you for this interesting question. While some studies mention that the original event has a return period of 70 to 100 years, we did not account for the return period for multiple reasons: 1) available return period estimates were derived quite a while ago (often under stationary assumptions) and might thus not represent the effective return periods. 2) as we analysed future scenarios as well, it was much more comprehensive to analyse and compare flood damage information than flood risk estimations given that we had no time or resources to analyse changes of return periods over time.

- I can generally understand the use Google StreetView and estate-agent ads for assessing information on buildings. But was this information recent and accurate -

how was this evaluated? Since some of the co-authors are actually based in Venice, this seems like something that could be easily done in the field.

We thank the reviewer for this in-depth question. Timeliness and accuracy of used data was evaluated together with experts (the co-author based in Venice, as well as Dr. A. R. Scorzini who is an expert on damage modelling in Italy). We also implicitly assumed that derived information (mainly physical appearance of exterior and interior of buildings) is rather static over time.

- Line 245: how was it detected, where from?

These observations were detected using GooleMaps StreetView. It was observed in multiple districts (i.e. San Marco, Dorsoduro, San Polo and Cannaregio). It was only observed at houses that seemed to be used for shopping/economic activities. These observations were discussed and confirmed with the local co-author.

- Although generally well written, the manuscript needs to be checked for some small language errors and some inconsistencies in the use of some terms (e.g. exposure)

Thank you for this comment. We reviewed the manuscript thoroughly and corrected spelling errors. The definition of exposure terminology has been adjusted in line 24 as follows: "*According to the IPCC, flood risk is defined as the combination of a specific hazardous flood event, elements (i.e. infrastructure, people, livelihoods, environment, and cultural, social and economic assets) which might be exposed to a hazard in a certain area, and the vulnerability of these elements, meaning predisposition to be adversely affected \citep{Field.2012}.*"

---

## Author Response (AR2)

Dear Editor, first we want to thank you for your constructive, in-depth and clear questions and comments. Below you can find our answers point-by-point to your comments. We also highlight (where possible) the part of the original submitted manuscript that have been modified to address your comments. To facilitate the reading, we added our responses to your comments in red. If changes to the text are proposed, changes are underlined.

1. Please ,clarify what is new in your study. Methodologically what is new? What is the relevance of providing estimates of damages considering only a single event in different operational configurations and SLR scenarios? I suggest that you advocate more connivingly in the manuscript the value and practical utility of your study

Thank you for this comment. We propose the following section being added to advocate more explicitly for the value of the study:

At the end of line 71: "*As such, no risk assessment framework is accessible that captures the flood dynamics or allow for a comprehensive adjustment of exposure and vulnerability due to urban developments for potential long-term use of such frameworks. Flood dynamics might be altered in future because of the operation of the MOSE barrier influencing the bathymetry and thus hydrodynamics of floods in the Venetian lagoon (Tognin et al. 2022).*"

At the end of line 78: "*The framework is tested using the second highest recorded flood event for which damage claim data have been collected and made available by the municipality. Those most-recent damage claim data were used to analyse and discuss the suitability of the framework by comparing these empirical data with the simulated flood damages of the framework.*"

Moreover, we propose the following addition in line 540: "*[…] in accordance with available damage claim data. While the use of a hydrodynamic model posed some numerical challenges and resulted in similar flood damage estimates than based on a bathtub model, the opportunity to integrate additional elements such as wave-effects or a 1D-flow path-component representing the sewage system in the low-lying city might allow for a more accurate flood hazard estimation beneficial for efficient flood risk management.*"

Finally, we propose the following addition in line 541: "*[…] limited knowledge about the system and damage processes. Various existing approaches and elements (hydraulics, damage model, interventions, sea level rise scenarios) were integrated to develop and test a novel approach to risk assessment for Venice. While the application focus of this study focuses on events with a single return period the framework can be easily used to consider other events (with other return periods) to come to a complete risk assessment in current and future conditions and for various interventions. Given the complexity of the system and the large numbers of possible interventions, it would be a study by itself to evaluate all the (combinations) of interventions (Berchum et al. 2019). Thus, in this paper we have focussed on the introduction of the framework and its illustration for a limited number of events and interventions.*"

2. Line 3 replace mitigate with reduce (see IPCC glossary for the meaning of "mitigation"). Further at different points you mention "exposure … characteristics from typical residential buildings". This concept is not clear. Please check the IPCC glossary for the meaning of exposure and revise the text (eventually delete the term "exposure")

Thank you for this comment. We replaced the word "mitigate" in line 3 and further revised the use of exposure characteristics. Accordingly, we propose changes of the following lines:

l.201f: Consequently, the chosen model was selected with special care to allow for an inclusion of differing  vulnerability characteristics.

l.212f: INSYDE is a multi-parametric model adopting 23 parameters to describe hazard,  and vulnerability characteristics of buildings

l.217f: The INSYDE model also makes use of building-type categorization to account for differences in the  vulnerability characteristics between typical buildings in a study area.

l.257f: […] was added to account for observed differences in the  vulnerability characteristics from typical residential buildings […]

l.464: […] preparation and protection measures on structure and neighborhood level, as well as other vulnerability characteristics, in Venice […]

l.562: Also, a better understanding of the spatial distribution of protection measures and other damage mediating characteristics […]

3. Line 97 replace "ever recorded " with a more precise statement, such as that this event is the second highest water height since the beginning of the 150 year long instrumental record, (ref: Lionello et al., 2020, already cited by the authors)

Thank you very much for this comment. We adjusted the sentence in line 97 as follows: "*On 12 November 2019, the second highest sea level since the beginning of measurements (1872) flooded the old-town of Venice and other parts of the Venetian lagoon.*"

4. Line 159-165. comment to which extent the lack of interaction among nested models may lead to inconsistencies in flow velocity and water level among the different areas and whether this is relevant for estimating the damages

We agree with your suggestion and propose adding the following sentence in line 166: *[…] not exchange information among each other but were run independently. Inconsistencies in flow velocities and water levels due to the lack of interaction between the sub-models were neglected given that most interaction was assumed to occur through the canals which were sufficiently captured already in the parent model using a resolution of 2.6m within the city.*

5. Line 322-331 and table 8. To which extent the lack of c3dfm result for the "Castelo" sub-model are relevant for table 8. In other terms what is the weight of the Castelo's results in the damage costs and claims? Please comment and add a sentence explain this

Thank you for this comment. While we can give an indication of the effect of lack of d3dfm results in "Castello" for the immediate response claims, the more extensive claims were provided by the city of Venice only aggregated preventing any indication of the effect of different water depth estimates. We propose the following addition in line 324: "*A total of 94 immediate response claims (2.5 % of immediate response claims, amounting for 656,264 EUR in claim volume) were located in the sub-model "Castello". As indicated before, we used flood depth estimates from the bathtub model resulting in minor effects on the structure-wise results. "*

And also in line 461: "*[…] to address the mentioned limitations of the framework. While the structure-wise analysis of immediate response claims allowed for a comparison of the bathtub model and hydrodynamic model, the high-aggregation level of all damage claims in*

*combination with the numerical challenges in the hydrodynamic sub-models, did not allow to confirm findings of this comparison."*

6. Line 429 clarify what is meant for "proper calibration"

We propose the following rephrasing: *"a fully functioning hydrodynamic model may add additional benefits to the flood risk assessment framework as it can account for (changing) physical characteristics explicitly, allow for a calibration based on flood depth information, and incorporate additional flow path-components such as a 1D sewage system, which might lead to different flooding patterns"*

7. Line 420-430 this paragraph suggests that the feasibility of using an hydrodynamic model has been hampered by numerical problems. Further, the sentence 425-426 suggests that in general there might be a limited advantage running a fluid dynamic model with respect to a bathtub model. To which extent it is important to use a dynamic model with respect to a bathtub model? This study provides little evidence for this and the text of the "conclusions" section is not clear about this.

We propose the following addition in line 540: "*[…] in accordance with available damage claim data. While the use of a hydrodynamic model posed some numerical challenges and resulted in similar damage estimates than based on a bathtub model, the opportunity to integrate additional elements such as a 1D-flow path-component representing the sewage system in the low-lying city might allow for a more accurate hazard estimation beneficial for efficient flood risk management."*

8. Line 246-253 please include information on how the building information assessed from Google streetview was evaluated and to which extent this add uncertainty to the estimates of the damage

We propose the following additional sentence at the end of line 248: *[…] at ten random locations in different districts of the old-town. At each of the random locations, we regarded house fronts on both sides up to a distance of 50 to 250m in various directions from the starting point. In this way, we obtained information regarding an estimate of 300 buildings. Length information were estimated based on expert judgment, available scales (e.g. door dimensions). In this way, a first-order estimation of building information was obtained in absence of available statistical data. These building characterstics were confirmed with local inhabitants.*

**Additional alterations:**

- Based on feedback we received from the editorial board, we adjusted Table 5 to avoid the use oft he colours red and green.
- We have provided some updated references in some part of the paper
- We have updated the affiliation of one of the two authors (Julius Schlumberger, Manuel Andres Diaz Loaiza)